# Citclops: A next-generation sensor system for the monitoring of natural waters and a citizens' observatory for the assessment of ecosystems' status

Luigi Ceccaroni[1]*, Jaume Piera[2], Marcel R. Wernand[3†], Oliver Zielinski[4,5], Julia A. Busch[4,5,6], Hendrik Jan Van Der Woerd[7], Raul Bardaji[8], Anna Friedrichs[4,9], Stéfani Novoa[1], Peter Thijsse[10], Filip Velickovski[11], Meinte Blaas[12], Karin Dubsky[13]

1 Earthwatch, Oxford, United Kingdom, 2 ICM, Consejo Superior de Investigaciones Científicas, Barcelona, Spain, 3 Royal Netherlands Institute for Sea Research, Den Hoorn, Netherlands, 4 Institute for Chemistry and Biology of the Marine Environment, University Oldenburg, Oldenburg, Germany, 5 Marine Perception Research Group, German Research Center for Artificial Intelligence, Oldenburg, Germany, 6 Common Wadden Sea Secretariat, Wilhelmshaven, Germany, 7 Institute for Environmental Studies, Vrije Universiteit Amsterdam, Amsterdam, Netherlands, 8 Marine Technology Unit, Consejo Superior de Investigaciones Científicas, Barcelona, Spain, 9 Federal Maritime and Hydrographic Agency, Hamburg, Germany, 10 MARIS BV, Nootdorp, Netherlands, 11 Quantium, Sydney, Australia, 12 Rijkswaterstaat Water Transport & Environment, Lelystad, Netherlands, 13 Department of Civil, Structural and Environmental Engineering, University of Dublin Trinity College, Dublin, Ireland

† Deceased.
* lceccaroni@earthwatch.org.uk

**Data Availability Statement:** Through the interface at [http://www.citclops.eu/search/welcome.php] users are able to easily download all data. All data

## Abstract

The European-Commission—funded project 'Citclops' (Citizens' observatory for coast and ocean optical monitoring) developed methods, tools and sensors, which can be used by citizens to monitor natural waters, with a strong focus on long-term data series related to environmental sciences. The new sensors, based on optical technologies, respond to a number of scientific, technical and societal objectives, ranging from more precise monitoring of key environmental descriptors of the aquatic environment (water colour, transparency and fluorescence) to an improved management of data collected with citizen participation. The sensors were tested, calibrated, integrated on several platforms, scientifically validated and demonstrated in the field. The new methods and tools were tested in a citizen-science context. The general conclusion is that citizens are valuable contributors in quality and quantity to the objective of collecting, integrating and analysing fragmented and diverse environmental data. An integration of these data into data-analysis tools has a large potential to support authoritative monitoring and decision-making. In this paper, the project's objectives, results, technical achievements and lessons learned are presented.

are also available from the dataset with DOI: 10.5281/zenodo.3497440.

**Funding:** The funders of this study provided support in the form of salaries for authors LC, JP, MRW, OZ, JAB, HVDW, RB, AF, SN, PT, FV, MB, KD, but did not have any additional role in the study design, data collection and analysis, decision to publish, or preparation of the manuscript. The specific roles of these authors are articulated in the 'author contributions' section. More specifically, all authors (LC, JP, MRW, OZ, JAB, HVDW, RB, AF, SN, PT, FV, MB, KD) received funding from the European Union's FP7 research and innovation programme under grant agreement No 308469 'Citclops'. LC received funding from the European Union's Horizon 2020 research and innovation programme under grant agreements No 824711 'MICS' and No 824580 'EU-Citizen.Science'. The opinions expressed in this study are those of the authors and are not necessarily those of the Citclops, MICS or EU-Citizen.Science partners, or the European Commission.

**Competing interests:** Two of the authors are currently employed by commercial companies: MARIS BV and Quantium. Peter Thijsse is employed by the commercial company MARIS BV. Filip Velickovski is employed by the commercial company Quantium. The authors have declared that no competing interests exist. The commercial affiliation of PT and FV did not play a role in this study, in the decision to publish, or in the preparation of the manuscript. Also, this commercial affiliation does not alter authors' adherence to PLOS ONE policies on sharing data and materials.

# Introduction

Citizen science is defined as work undertaken by civic educators and scientists together with citizen communities to advance science, foster a broad scientific mentality, and/or encourage democratic engagement, which allows people in society to join the debate about complex modern problems [1]. This definition shifts the focus from the action-oriented, data-centred point of view of *collect*, *participate* and *contribute* towards how science and society should respond to a call for *openness*, *inclusiveness*, *responsiveness*, *democratic engagement*, *consultation*, *dialogue* and *commons*. The definition reflects the following values:

- supporting and advancing scientific research (Data collected in citizen-science projects can be used to complement data collected in traditional ways by scientists and governments.) [2];

- public engagement in scientific discourse;

- public engagement in informing policy at various levels, from local to international;

- desire to achieve a particular environmental, social or policy outcome;

- increased capacity to respond to community needs, such as concerns about water quality or access to scientific information;

- enhancing lifelong learning/education about the scientific process, and complex modern problems.

In the European-Commission—funded citizen-science project *Citclops* ("Citizens' observatory for coast and ocean optical monitoring", funded during 2012–2015), about 3000 members of the public partnered with scientists and engineers to solve complex problems through participating in all of the following processes: formulation of questions and experiments; collection and analysis of data; and interpretation, use, and publication of results. Encompassing citizen sensing, community science, and related approaches, Citclops's citizen science also benefited public participants by providing opportunities to learn, addressing questions of concern to the participants and their communities. Two websites and two apps have been created to facilitate this participation, which will be described in detail in the rest of the paper:

- [**citclops.eu**];

- [**eyeonwater.org**];

- *EyeOnWater—Colour* **app**, that allows to measure the colour of water;

- *KdUINO Remote Control* **app**, that allows to control an instrument to analyse water transparency.

Although terrestrial citizen-science projects are more visible, in recent years thousands of volunteers have actively participated in marine-research activities [3–4] and in freshwater-research activities [5]. In the Citclops citizen-science project, civic educators, among whom the authors of this research, were specifically concerned about data related to the environmental status (including pollution) of fresh and saline waters (limnological and oceanographic data). A widely adopted, scientific approach to assess the environmental status of water bodies is by measuring their optical properties, such as the colour of the water, its transparency or clarity, and its fluorescence.

These data can then be used to complement and improve the reporting of data related to the *United Nations* (UN) *Sustainable Development Goals* (SDGs). For example, essential to SDG 14 is the target of achieving substantial reductions in marine pollution, including nutrient

pollution and marine debris in coastal waters (SDG Indicator 14.1.1). Eutrophication is increasing in coastal waters, and UN Environment—the custodian agency responsible for this indicator—recommends the combined use of remote sensing and citizen science for large-scale monitoring with validation by citizens. Mobile applications such as Citclops's EyeOn-Water—Colour enable volunteers to contribute data on the colour of coastal waters, which serves as a simple and accessible baseline indicator for eutrophic trends that can be used in tandem with remote-sensing data [6].

The apparent *colour* is a result of substances that are either suspended or dissolved in the water column [7]. There are three main components besides the water itself affecting the water colour in natural environments via their specific scattering and absorption properties: organic particulate matter (mainly, phytoplankton); inorganic particulate matter (mainly, chalk and mineral parts); *coloured dissolved organic matter* (CDOM, also denoted as Gelbstoff). Colour's unit of measurement is based on the Forel-Ule colour comparator scale [8]. Over a century ago, not only scientists collected oceanographic data, but also merchant sailors. Forel-Ule observations are a good example of such globally collected data, the first ones of which date back to 1889 [9].

*Turbidity* (the opposite of clarity) is the combined effect of scattering and absorption of light, mainly on particulate matter originating from biological activity (phytoplankton and zooplankton debris), and on suspended inorganic solids. The simplest way of measuring it is by using a white disk of 30 cm in diameter (the Secchi disk) [http://www.secchidisk.org/] or alternatively a quadrant-divided black-and-white disk of 20 cm in diameter (the Whipple Disk). The disk is lowered on a length-marked cable until it disappears from view, and then raised until it just reappears; the latter measurement is a proxy of clarity (known as Secchi depth). It varies with the wind, season, vegetative growth, faunal activity, decomposition of vegetation, and sediment resuspension and flocculation [10].

*Attenuation of the light in water* (the opposite of transparency) relates to a measure of the depth of light penetration into the water [10], which is relevant for underwater primary production. It depends on the amount of the following components: CDOM, and organic and inorganic particulate matter. Light, during its propagation in the water, attenuates due to absorption and dispersion by molecules and particles until completely disappearing. In citizen science, the most used instrument to measure the transparency is again the Secchi disk. Advantages and disadvantages related to the use of the Secchi disk are described in this paper, together with a novel technology to measure the transparency, developed within Citclops.

*Fluorescence* is important to detect substances such as particulate matter (e.g., phytoplankton) and dissolved matter (e.g., coloured dissolved organic matter) [11–12]. The development of novel technologies, methods and sensors for fluorescence measurements is one objective of the research described in this paper. Specific target constituents for fluorescence measurements in this study are *coloured dissolved organic matter*, phytoplankton, as well as hazardous substances that belong to the group of *polycyclic aromatic hydrocarbons* (PAHs).

Citizens have been enabled to carry out participatory monitoring of these characteristics, based on low-cost devices, achieving two main goals: (1) to expand the range of monitoring possibilities; and (2) to raise the citizen engagement with environmental issues. *Engagement* of citizens with other stakeholders (science and technology researchers, and decision makers) in a *citizens' observatory* such as Citclops is a process, which evolves from initial stages of maturity (e.g., widespread water colour and transparency misconceptions) to more developed understanding. In the early stages, Citclops' researchers consulted citizen communities as they planned, designed and started to implement the project. It is critical to note that the participatory engagement of stakeholders was perhaps the most distinctive component of the planning and development of Citclops, which led to its growth and adoption without advertisement.

The key is an architecture of participation, with which a citizen-science project does not need to promote itself because, unlike their non-participatory counterparts, it connects people directly to engage in meaningful activities. MARIS, the enterprise which manages some of Citclops's output results, makes users productive and engaged by letting them share water-related data with any other user worldwide for free.

Before the advent of citizen science, building an international monitoring infrastructure and maintaining it required a colossal pool of capital. With citizen science, this is now less expensive; nevertheless, it has to be considered that the European Commission has dedicated several million euros to get it started, through projects like Citclops.

Over the three-year period of the Citclops project's execution (between October 2012 and September 2015), new methodologies, instruments and measurement techniques have been developed and exploited in the framework of citizen science and the observation of natural-waters (rivers, lakes, coasts and oceans) (see [www.citclops.eu]). A total of 11 partners from five countries (Spain, The Netherlands, Germany, France and Ireland) were involved, including academic and educational institutions, technology centres, industry and end-user organisations, with wide experience of service deployment to create demonstration environments that can be used directly by a number of different target audiences to: increase community participation; to create environmental awareness at selected sites; to collect scientific data, with a focus on the marine environment; and to facilitate visual as well as instrumental water-quality monitoring of natural waters.

The project included the design of electronic devices to allow measurements of colour, transparency and fluorescence of natural waters, with the following objectives:

- to enable citizens' participation in acquiring environmental data of natural-waters quality through the use of existing devices, such as smartphones, used as sensors (This includes the development of apps, friendlier and more flexible user interfaces, and social-networking capabilities to connect citizens and their associations among themselves and to researchers.);

- to develop improved, low-cost sensors and systems for geo-located monitoring of water colour, transparency and fluorescence, allowing for the analysis of spatial patterns;

- to provide recommendations in sectors such as fisheries and health, interpreting collected data through data-analytic techniques;

- to disseminate interpreted information to three kinds of users: citizens (individuals and associations), policy makers (e.g. local administrations) and researchers.

In the following sections, the methods and main accomplishments of the Citclops project are presented, structured according to the three optical measurement approaches, namely water *colour*, *transparency* and *fluorescence*, and to the engagement of citizens. Finally, conclusions and recommendations are presented.

## Methods

### Colour

Observations of the colour of natural waters, performed through a colour comparator scale, date back all the way to the late 19th century. Just as satellites monitoring our sea, monitoring performed through the colour comparator scale, the so-called *Forel-Ule* (FU) scale has covered lakes, seas and oceans since 1889 and took place all over the world. The usefulness of the FU-scale observations has been proved over the years. Recently, the FU dataset, containing world-wide-collected data for over a century, has been analysed to determine possible trends in climate [13]. The reintroduction and use of the FU observation method would facilitate climate

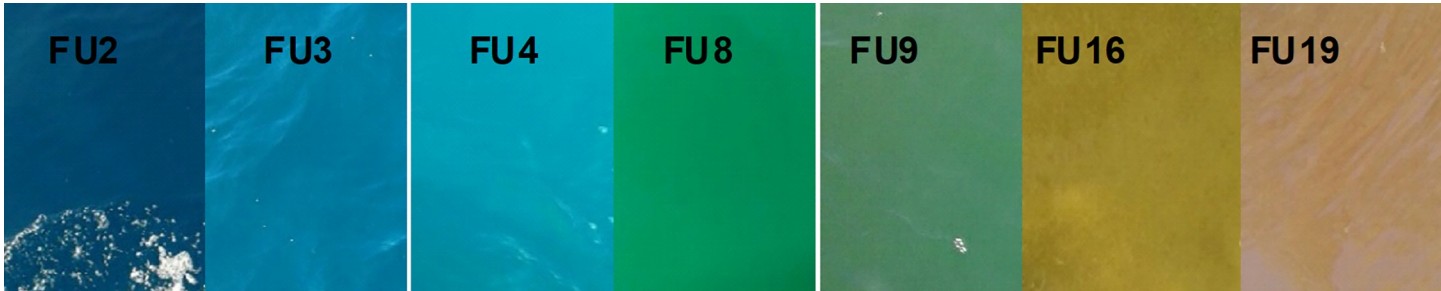

**Fig 1. Examples of images sent by the public, with the correspondent FU index value.**

research, and the determination of short-term changes of the aquatic environment. Accordingly, the Citclops project contributed to it through citizen science, with the help of the public, and the development, manufacturing and distribution of a new type of easy-to-make Forel-Ule scale. Additionally, a smartphone colour app has been developed to facilitate public involvement in worldwide collection of FU data.

The complete FU scale was designed in two steps between 1890 and 1892. The Swiss researcher François-Alphonse Forel used 11 colour-tones covering the blue to green waters to study the colour of Lake Geneva. Two years later the German Wilhelm Ule extended the scale by covering green to brown waters by adding ten extra colour-tones. So, the scale is composed of 21 colours, going from indigo blue to 'cola' brown, through blue-green, green, and yellow colours (Fig 1). The scale has been fully described by Wernand and Van der Woerd [14] and by Novoa et al. [15–16]. Although the FU scale more or less fell into disuse after the introduction of hyperspectral optical sensors, the value of the data collected through the use of the scale should be realised. Historical FU observations of the colour of the sea are the only available data that can tell us something, although indirectly, about the amount of chlorophyll (a proxy of phytoplankton abundance) in the sea over the long term (with respect to the relatively short period over which satellite ocean colour has been collected). The FU observation method consists of a simple determination and classification of the colour of natural waters by comparing the colour of the water under investigation with the colour palette of the FU scale (21 coloured liquids in vials) (Fig 2).

This system of colour comparison works well as the human eye can accurately match colours when viewed simultaneously. The colour measurement with an FU scale should be done with a Secchi disk, positioned at half the Secchi disk depth, as background, but methods have been developed within the Citclops project to assess the water colour without the use of a disk, and without the need of human-eye matching. The authors accomplished the determination of the intrinsic water colour using either a FU scale made of coloured plastic filters (the Modern FU-scale) and human assessment or a smartphone picture and an algorithm.

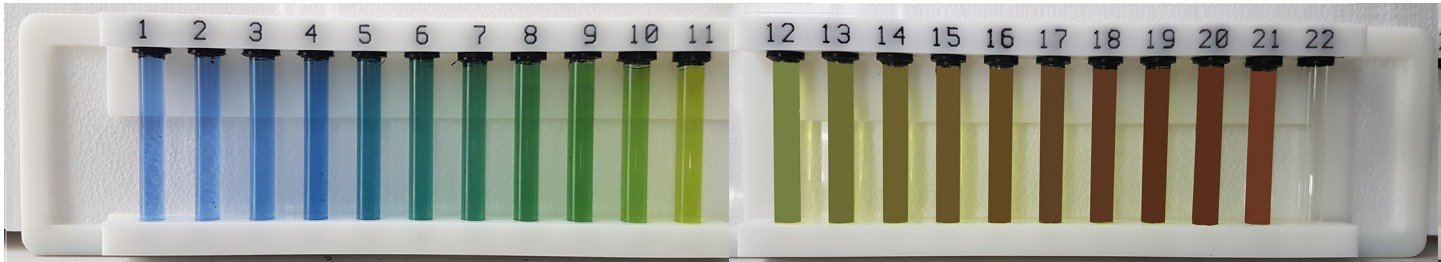

**Fig 2. The classic Forel-Ule scale: 21 coloured liquids in vials.**

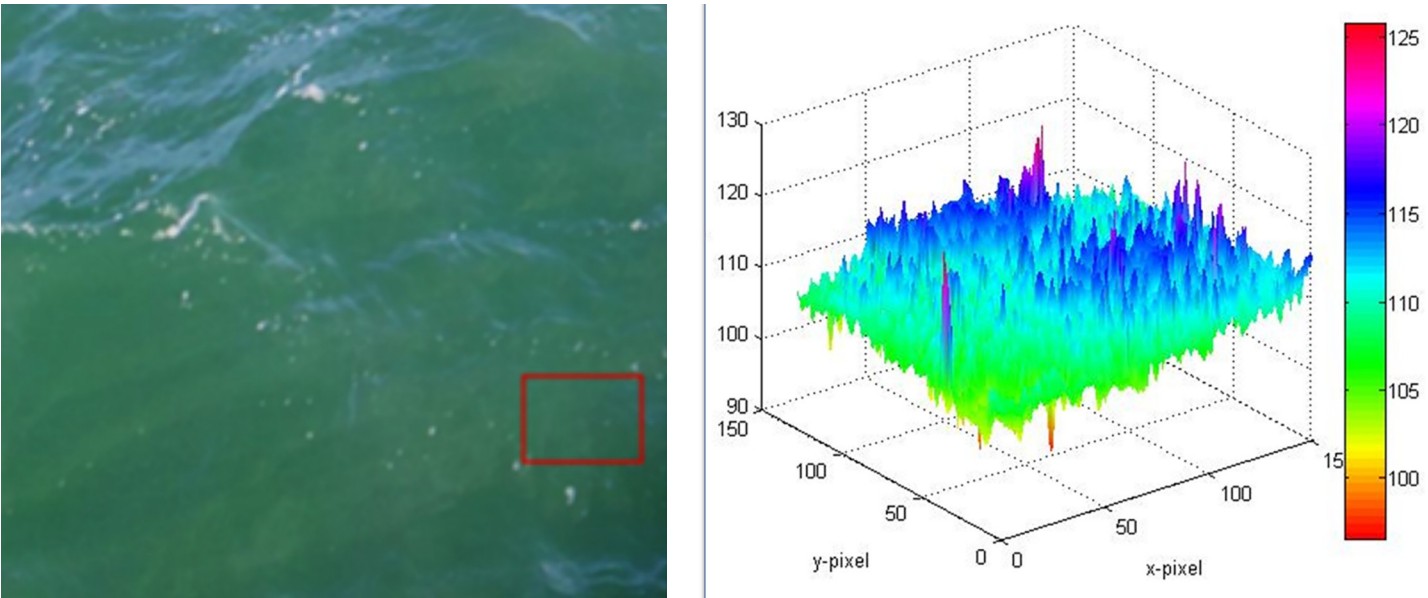

**Fig 3. Water colour at the micro scale with a resolution of approximately 1 square centimetre (right).** The image of the North Sea derived from a smartphone camera (left) is converted to the hue angle with WACODI software.

This algorithm (WACODI), developed within Citclops, is able to classify a valid image of surface water into one of the 21 FU colour classes. The algorithm has achieved this result after having been trained by human-supervised classification through the "Citclops—Citizen water colour monitoring" app [17], in which users manually provided the FU value. The algorithm locates, in a citizen-supplied image (or any image of water surface), a subsection that contains water, then computes the closest FU colour value of the subsection. To compute the FU colour value, it uses colourimetry and an adaptation of the methodology employed by Wernand et al. [13] to estimate the FU index using satellite imagery, in this case applied to digital images from smart phones and low-cost digital cameras. See Fig 3 as an example.

For colour computation, the following validation activities have taken place:

a. The FU scale as provided by the observer in the "EyeOnWater-colour" app (see section Citizen water-colour monitoring apps) is compared to an FU scale that is derived with the WACODI algorithm [17] at the server.

b. The WACODI algorithm was validated with state-of-the art above-water hyperspectral sensors in a field campaign that covered a wide range in natural waters, from lakes to oceanic waters.

c. The protocols and algorithms to derive the FU scale from the MERIS satellite observations were validated with in-situ measurements. The results for many different natural waters are described by Wernand et al. [13]. A more extensive validation campaign in lakes near the Ebro has been published by Busch et al. [18].

Colour, together with clarity, is one of the most apparent characteristics to the human eye when observing natural water. In general, the apparent colour is a result of substances that are either suspended or dissolved in the water column. There are three main natural components, besides the water itself (Oligotrophic ocean water has an indigo colour.), affecting the water colour and clarity in natural environments (Fig 4):

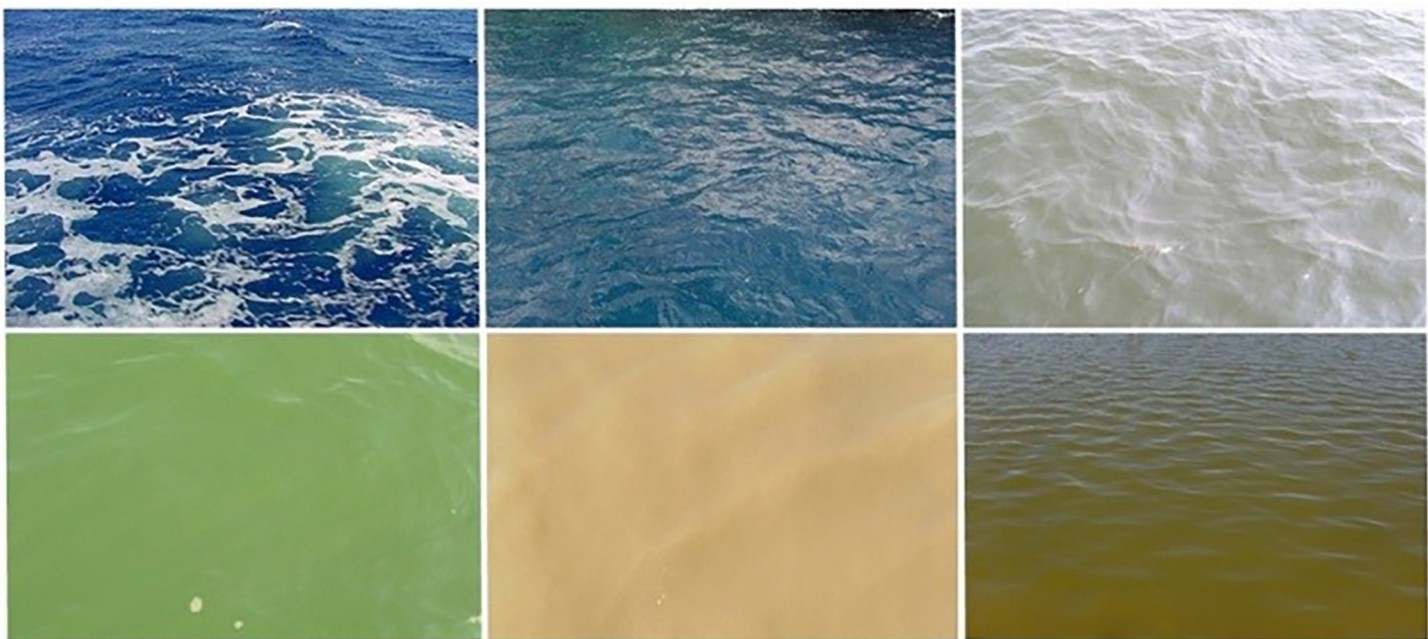

**Fig 4. Examples of six differently coloured sea areas.** Top, from left to right: Central Atlantic, Central North Sea, and Coastal North Sea. Bottom, left to right: Coastal North Sea during algal bloom, Wadden Sea with re-suspension of sediment and a coastal outlet dominated by coloured dissolved organic matter (photos courtesy of Bert Aggenbach, Annelies Hommersom and the authors).

1. organic particulate matter (mainly, **phytoplankton**) (generally green);

2. inorganic particulate matter (mainly, chalk, mineral parts and **fine soils**)(milky, grey, dark);

3. **CDOM** (also denoted as Gelbstoff) (yellow to brownish).

Some natural phenomena can change water colour, and a particular colour does not necessarily mean that the water is of bad quality. For example, the spectral reflection of natural waters might be modified by the fluorescence signal of algae, CDOM and microbes in the water. In natural waters, fluorescence by algae is a signal located near 685 nm and rather weak compared to the full algal absorption over the visual range. The quantum efficiency is of the order of 1%. Also, the colour matching function for the red band near 685 nm is very low, which implies that a small addition in reflection will have marginal impact on the red colour and thereby the observed FU scale. Therefore, interference by fluorescence in the colour observation might be present, but is ignored for the moment. To determine the ecological status of surface waters, the quantification of the presence of pollutants, such as accumulations of plastic debris, will be also necessary.

The different FU colour numbers correspond to the following types of water bodies:

- **Indigo blue to greenish blue (high light penetration)** (1–5 FU scale). These waters often have low nutrient-levels and low production of biomass. The colour is dominated by microscopic algae (phytoplankton).It is typical of areas of open sea.

- **Greenish blue to bluish green** (6–9 FU scale). The colour is dominated by algae, but dissolved matter and some sediment may be present. It is typical of areas of open sea.

- **Greenish** (10–13 FU scale). Coastal waters usually display high nutrient and phytoplankton levels, and also contain minerals and dissolved organic material.

- **Greenish brown to brownish green** (14–17 FU scale). These waters usually have high nutrient, phytoplankton, sediment and dissolved organic matter concentrations. They are typical of near-shore areas and tidal flats.

- **Brownish green to cola brown** (18–21 FU scale). These waters have an extremely high concentration of humic acids, which are typical of rivers and estuaries.

The authors also produced and published several videos on the colour of the sea for the general public, to be used in citizen science (e.g., DOI: 10.5281/zenodo.3668848 or DOI: 10.5281/zenodo.3668850).

## Transparency

One parameter to quantify water transparency is the *diffuse attenuation coefficient* ($K_d$), which is estimated by measuring the decrease of *downwelling irradiance* with depth. $K_d$ is highly correlated with phytoplankton concentrations and is often used in water environment studies. $K_d$ is also related to remote sensing data because about 90% of the light received by remote-sensing sensors (the reflected light from the water column) comes from a surface layer of $1/K_d$ depth [19], making $K_d$ a critical parameter to correctly interpret remote sensing data.

To date, there are three main ways to measure $K_d$:

- *With radiometers*. The radiometers provide high-quality values, but the cost of the equipment is not affordable for many research groups.

- *With remote sensing*. Remote sensing via satellite measurements is a powerful tool to achieve good spatial coverage. However, remote sensing data presents some limitations in coastal zones and small lakes. When remote-sensing pixel size is around hundreds of meters or some kilometres, the pixels of coastal waters and small lakes are "contaminated" with ground areas, and cannot be used. In addition, clouds and low sampling-frequency reduce the availability of remote-sensing data.

- *With the Secchi disk*. The Secchi disk is a typical citizen-science instrument that allows measuring the Secchi depth from which $K_d$ can be estimated. The tool provides low precision measurements, and requires active human implication (one trained person has to be present for each measurement).

In Citclops, a low-cost moored system (KdUINO) that measures $K_d$ has been developed [20], whose electronics are based on an open-source hardware platform. The KdUINO contains a data-logger and several quasi-digital optical sensors from which it measures the light intensity. Post-processing analysis allows calculating the $K_d$ parameter by using the relative irradiance values obtained from the different sensors. The simple structure of the system combined with the low-cost electronics (under 100 USD) converts this instrument into a suitable *do-it-yourself* (DIY) research tool for citizen contributors, laboratories and environmental-monitoring companies with the need of cost-effective optical measurements of water quality ($K_d$ estimations) with a high spatial and temporal resolution. The authors produced and published a video tutorial on how to build a KdUINO [https://vimeo.com/115469736].

## Fluorescence

Unlike apparent characteristics of natural water, such as its colour and transparency, fluorescence is not usually visible by eye when looking at it. This is due to the weakness of the signal,

when compared to the backscattered light. While colour and transparency capture bulk properties of substances in water, fluorescence is a specific measure to retrieve water quality parameters in a selective way.

Fluorescence of substances is excited by exposure to light. Different fractions of light (colours) specifically excite fluorescence of different substances. The colour of fluorescence is at a lower energetic fraction (a different colour) than the excitation light. As an example, excitation of *"chlorophyll-a" algal pigment* (Chl-a) with blue light (higher energy) induces emission of red light (lower energy) as fluorescence, while blue *fluorescence of CDOM* (fCDOM or FDOM) is excited by *ultra-violet* (UV) light. The *fluorescence quantum efficiency* ($\varphi$) of phytoplankton is defined as the ratio of moles photons emitted as fluorescence divided by the moles photons absorbed by the pigments [21]. Measured values of $\varphi$ for *chlorophyll-a* (Chl-a) are in the order of 1% [21–22]. The pure fluorescence signal of Chl-a is emitted over a range of wavelength and the signal can be described by a Gaussian curve with a maximum height at 682 nm [7] in case of *in vivo* measurements. In case of Chl-a, the strength of the fluorescence signal is in first approximation dependent on the concentration of photo-active pigments in the sample and the intensity of the excitation source. However, other mechanisms can impact the quantum efficiency $\Phi$, like:

a.  photochemical quenching processes;

b.  non-photochemical quenching processes [23];

c.  temperature;

d.  photoinhibition in case light excitation is higher than saturation of photosynthesis [7];

e.  cell physiology [21]; and

f.  photo adaption of the algae.

Photo adaption occurs in various long-term and short-term processes like photo acclimation on time scales of seconds to days including day-night-circle resulting in biochemical, biophysical and biological changes and intra-cellular self-shading called package effect [24]. Also, for fCDOM several factors influence the fluorescence signal:

1.  chemical structure;

2.  presence of scattering particles; and

3.  protein-like and humic-like components [25–26].

Considering all signal disturbing processes, it is clear that measuring fluorescence is a challenging task especially in water samples without any treatment. Nevertheless, a principle frequently applied in commercial in situ measurements is that the signal can be locally and temporarily (linearly) coupled to the concentration of water constituents [11–12].

State-of-the-art sensor systems take advantage of these specific excitation-emission pairs for the selective measurement of different substances. A wide range of portable and robust fluorescence sensors is commercially available, each one with a light source for excitation (usually an LED, in few cases a Xenon flash) and a detector (usually a photodiode, in few cases a photomultiplier) to record fluorescence. Both are limited to appropriate excitation and emission wavelengths to capture components in water by means of high-quality filters. These combinations and principles are also the basis for Citclops fluorescence measurements. In Citclops, major target parameters for fluorescence measurements were *Chl-a* and fCDOM.

Two approaches to develop fluorescence sensory systems based on mobile devices were followed in Citclops:

a.  directly using the mobile-device sensor systems (SmartFluo); and

b.  using affordable external sensors integrated with the mobile device via smartphone applications (MatrixFlu).

In the first approach, fluorescence of a water sample in a cuvette (ex situ) with smartphones as sensor systems was measured, while, in the second approach, multiple parameters were measured directly in the water (in situ). The specificity towards the target substance promote both SmartFluo and MatrixFlu as relevant tools for environmentally engaged groups and water quality monitoring institutes [18–27].

For fluorescence measurement, validation was performed through parallel sampling and laboratory analysis following standard procedures at selected locations. Validation is explained in detail by Friedrichs et al. [28]: laboratory experiments of the SmartFluo show a linear correlation ($R2 = 0.98$) to the chlorophyll-a concentrations measured by reference instruments, such as a high-performance benchtop laboratory fluorometer (LS 55, PerkinElmer).

## Satellite observations

Monitoring of natural waters by space-borne sensors is operational since the launch of the NASA SeaWiFS sensor in 1997. Since that year satellite ocean colour imaging instruments have scanned the Earth with a spatial resolution in de order of 1 km and a temporal resolution (revisit time) of 3 days. The past and present satellite ocean colour imaging instruments are spectrometers that measure the backscattered sunlight in a limited number of narrow (≈10 nm) spectral bands in the visual domain. Traditionally, these observations are interpreted by scientist who calculate, on the basis of this spectral information, environmental descriptors such as the phytoplankton-biomass—proxy chlorophyll-a [29].

Within the Citclops project new pathways were investigated to couple the broad monitoring of satellites to localized (in space and time) in-situ measurements by citizens. Van der Woerd and Wernand [30] found that colour, expressed by the hue angle (α) or FU value, can be accurately and consistently derived from the space-borne sensors SeaWiFS, MODIS, MERIS and OLCI (Sentinel-3), thereby providing a natural connection among hyperspectral measurements, smartphone measurements and the historic colour observation of the 20th century.

In order to make this algorithm rapidly available for a wide audience, the hue-angle algorithm was also made available in the Sentinel-3 Toolbox [30], which is built on the Sentinel Application Platform [http://step.esa.int]. SNAP is a user-friendly platform for Earth-observation data processing and analysis and makes this information available to non-specialists. An example is shown in Fig 5.

The synergy between satellite and in-situ colour data was investigated by Busch et al. [31]. The in-situ observations of colour were directly compared to MERIS full resolution (300 m) ocean colour satellite data for the Ebro delta coastal zone in Spain with focus on the phytoplankton surveillance efforts in near-coastal areas and even shallow bays such as Alfacs and Fangar bays, that are economically and socially important areas (recreation and aquaculture). It turned out that spatio-temporal patterns of FU colour and hue angle of water were similarly displayed from ground and space, within their respective advantages and limitations:

• satellite data have superior temporal coverage and provide an excellent spatial background for the citizens to interpret and appreciate the value of their observation;

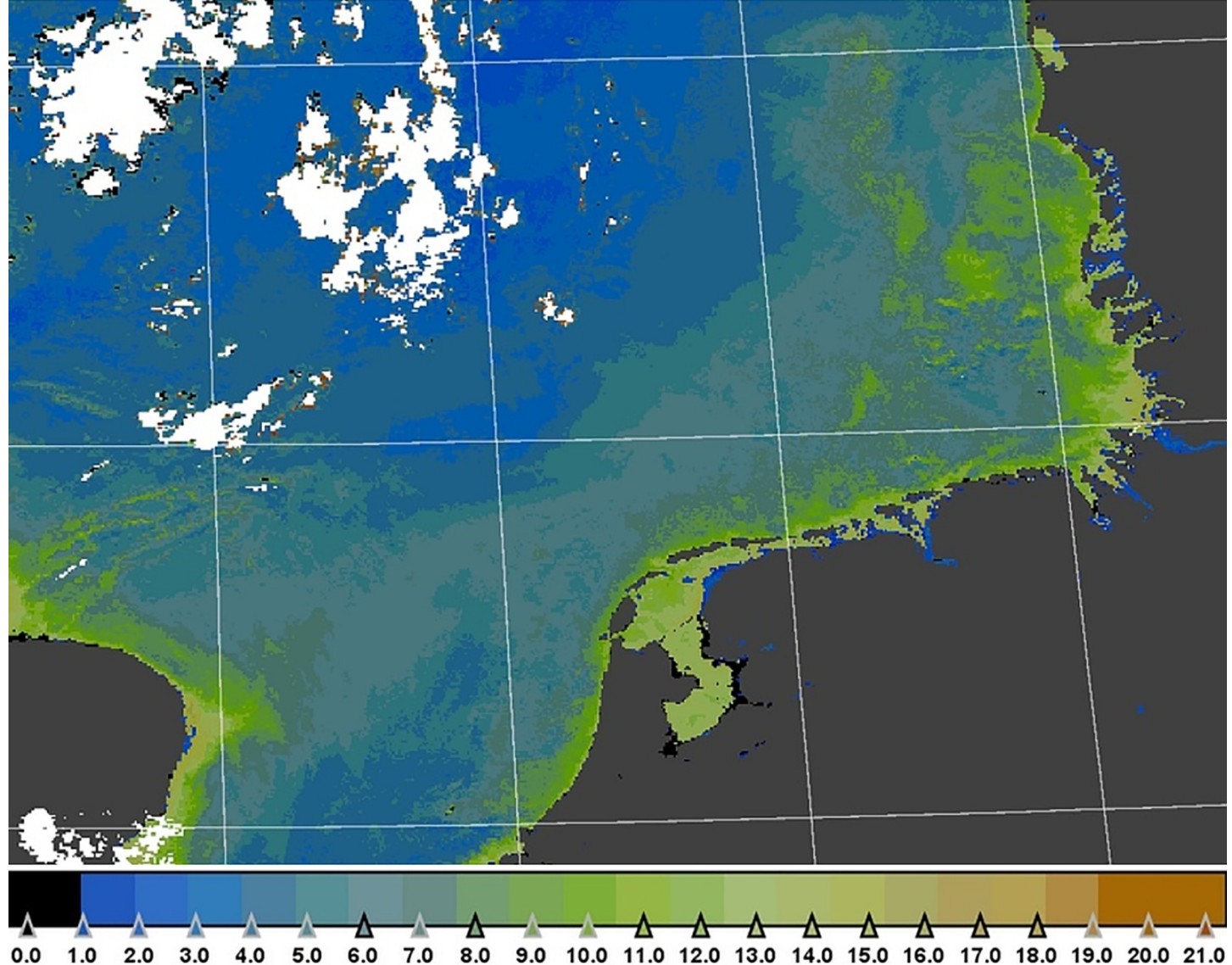

**Fig 5. Image of water colour at the macro scale with a resolution of approximately 1 square kilometre.** This image of the North Sea is derived from a MERIS observation and converted to the discrete FU values with SNAP software.

- satellite observations cannot be made in cloudy conditions while citizens can make observations any time;

- also, satellite products close to the coastline quickly deteriorate by impact of land/vegetation on the reflection spectra and citizen data can be used to complement or validate these satellite products.

## Data-quality control

Observations are valuable only if they closely reflect the true state of the environment and are fit for purpose. For example, the sighting and reporting of the presence of a "bird" carries limited information, while the reporting of a "barn owl" at a specific location and time can be

highly relevant. But which is the certainty that the species reported by a specific observer as "barn owl" is really a "barn owl" and not a "horned owl"? How can the quality of the data be checked? In science, the used methods are education, training and protocols, so that a scientist is properly trained to check in detail the conditions of an observation. Within Citclops the training, education and protocols were tested in multiple places with a large variety of people (see section "Education and contributor participation"), and a systematic analysis of potential disturbances in observation has been carried out [32]. In this systematic analysis, the key elements of perfect observations of colour, fluorescence and transparency are described in protocols and the potential disturbances are summarised. Good practices to detect errors and correct them are described. This includes the remote sensing observations.

The quality control procedures for observations of natural waters by standard marine instruments are already well developed and documented. These instruments have been already deployed and tested for a long period and that experience has helped to draft the protocols. However, within the Citclops project and in citizen-science projects in general, the instruments are under constant development. Also, due to the very participatory nature of the project, the instruments need to be low-cost and easy to be used by the public. This is a very different objective from the development of scientific instrumentation where everything is optimized to get an optimal observation and cost is no major issue.

Therefore, the authors formally described both the instrumental and environmental conditions that might have an impact on the quality of the observations. Based on this description, two data-quality protocols were defined: an a-priori quality protocol to guide the observation and an a-posteriori protocol to check the data after reception at the data server (see also [18]). The aim is not to reach the highest sensitivity for measurements, but to reach a consistent quality of citizen-science data, with clear documentation of associated errors or bias.

The new observation techniques described in this paper have been developed with the state-of-the art scientific insights and procedures. The idea behind this is that these citizen-science observations must have scientific merit. This goal was reached by detailed comparisons with standard laboratory equipment, field tests and publication in scientific journals. This is represented by connection "A" in Fig 2 of the paper by Busch et al. [18].

For colour, the Forel-Ule scale was characterized in the laboratory [15]. The construction of a plastic scale was developed under strict standardized illuminations conditions and a detailed protocol for the construction of this reference material has been published [16].

For fluorescence, standard fluorophores and chlorophyll solution have been used as reference. The method investigated and used in this paper was tested in a Carl von Ossietzky Universitat Oldenburg's laboratory in Wilhelmshaven. Field tests were performed at various sites. Additionally, two different scientific instruments were used to compare results: one instrument was a MicroFlu Chl-A from TriOS and the other one was a bench-top fluorescence spectrometer (LS 55, PerkinElmer). With respect to standard references, a dilution series of chlorophyll-a standard (5000 μg/L dissolved in 90% acetone) was used. No additional standard reference material was developed during test studies.

Mobile devices hold a number of sensors, e.g., *global positioning systems* (GPS) and clocks, and hence can be used to determine the crucial metadata: location and time. These data can be acquired automatically by the use of smartphone apps. In addition, smartphones hold many internal sensors that can guide the user to collect data, internally checking basic quality-control issues (angle to water surface, illumination conditions, and so on).

In the server, the second step of *data-quality control* (DQC) is directed at the processing of the database of collected observations and integration with other spatial information and satellite information. In Citclops, large volumes of data are collected with observations of natural waters all over the world under highly varying conditions (clouds, waves, solar elevation) and

by a wide range of observers, trained and not trained. DQC is as an important aspect when gathering samples collected from citizen-science initiatives as in traditional science. The Eye-OnWater—Colour app (see section "Water colour") allows any user to upload measurements without registration, or any experience in the field. Different field tests have been carried out involving the participation of users to evaluate their views on the operation and requirements of Citclops's apps. These users have been engaged to detect system failures and user-interface imperfections.

The DQC process uses the available information to decide to use the data for analysis in the Citclops process. For example, if the DQC process finds that the data contain too many errors or inconsistencies, then it prevents those data from being used in the decision support system (see section "Delivery of information to citizens, decision makers and researchers"), because they could cause disruption. For example, providing invalid measurements from smartphone sensors could cause invalid predictions of water colour. Thus, establishing DQC processes contributes to safe information usage.

Finally, a specific *citizen peer-to-peer quality control* mechanism has been implemented to filter and augment citizen observation data. With the purpose of detecting samples that do not correspond to a valid image of seawater, a process has been implemented to eliminate them. As can be seen in **Fig 6**, under the image corresponding to an observation there is a "Flag this!" button, which provides the possibility for users to flag the image in case they think that it is irrelevant within Citclops Data Explorer (see section "Delivery of information to citizens, decision makers and researchers").

## Results

Data (optical properties of water: colour, transparency and fluorescence) are collected by sensors; then they are stored in a database together with metadata; and are made available to users, either directly via the Citclops/EyeOnWater portals, or indirectly via overarching portals like EMODnet [http://www.emodnet.eu/] and GEOSS [http://www.earthobservations.org/geoss.php].

As Citclops involved participation of citizens and researchers in science, two website and two apps have been created to facilitate this participation:

- [**citclops.eu**] (Fig 7);

- [**eyeonwater.org**];

- *EyeOnWater—Colour* **app** (published in App Store and Google Play): an improved version of the *Citclops—Citizen water colour monitoring app* (also published in App Store and Google Play, but discontinued), that allows to measure the colour of water;

- *KdUINO Remote Control* **app** (published in Google Play, but discontinued): an app to control the KdUINO via Bluetooth. (KdUINO is used to analyse water transparency.)

As an example, the websites and the EyeOnWater—Colour app, not only facilitate the delivery of information to the public, but can be used by citizen scientists and researchers to look at the data collected. Shortly after the use of the EyeOnWater—Colour app for data collection, those data can be viewed at the Citclops/EyeOnWater websites. Fig 8 illustrates the about 5000 global observations available in the Citclops/EyeOnWater database and through GEOSS and EMODnet in June, 2018 (up from 1000 in June 2016). Dots correspond to images, inserted into the system by contributing observers (e.g., citizens, captains, and sailors) and including

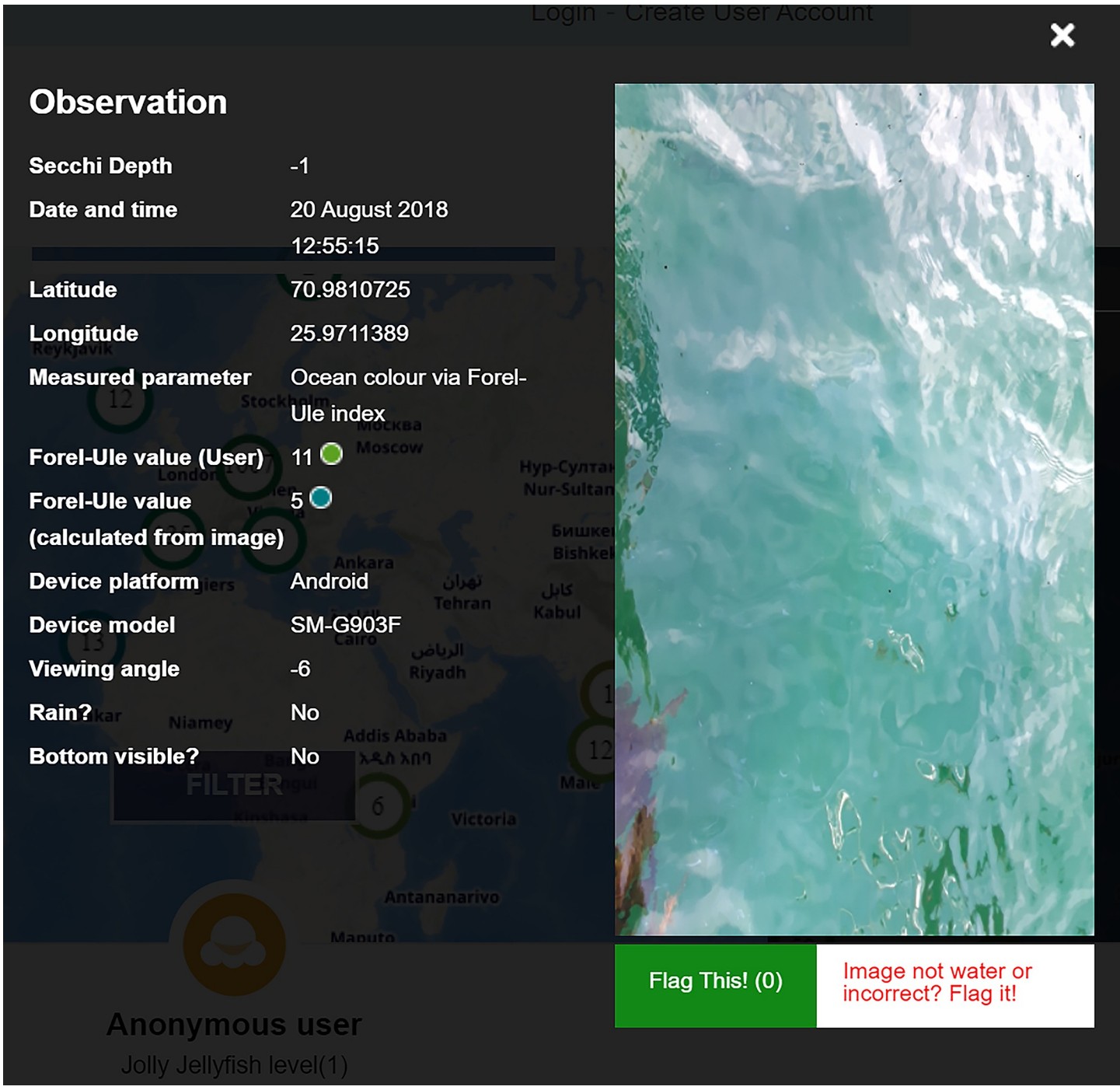

**Fig 6. Extended metadata pop-up.**

geo-location information. Clicking on one of the individual dots reveals the underlying information on observed and calculated colour, geographical position and date.

In the following sections specific results related to the three optical properties of water studied (colour, transparency and fluorescence) are presented.

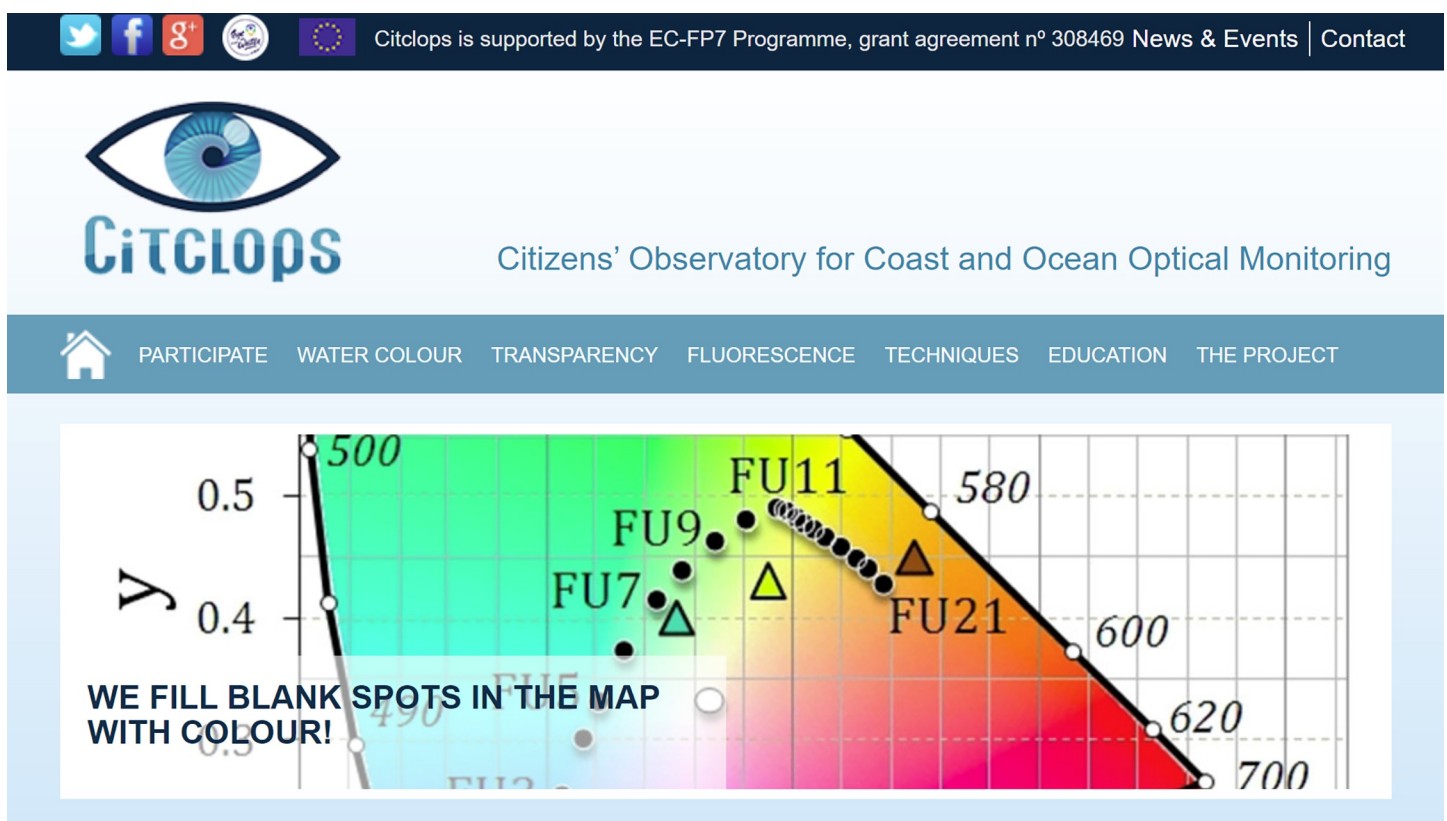

**Fig 7. Part of the Citclops website [www.citclops.eu].** Menu headers include the three main components of the project (colour, transparency and florescence) and guide the public towards specific information.

## Water colour

**The "Modern Forel-Ule scale".**   Framed within the Citclops project, a tool has been developed that can be employed by citizens to determine the colour of the water based on the FU colour comparator scale. As the 21 tubes of the original FU colour comparator scale partly contained carcinogenic liquids, new ways were investigated to mimic the scale colours. A scale, that accurately matches the FU colours and is composed of accessible and affordable materials, was developed by means of visual comparison and instrumental measurements. The new FU scale [15–17] is prepared using high-quality lighting filters (Lee, [http://www.leefilters. com]) and a frame made of durable plastic. The intention was to create a colour comparison tool that could be used by anyone willing to participate in environmental monitoring, providing an easy method to record not only the colour, but also the clarity of natural waters, since this scale needs to be used together with a Secchi disk (Figs 9–11). This tool has also been developed as a validation tool for the newly developed Citclops smartphone app, i.e. to validate the FU observation as performed with a smartphone app by comparing the value of FU selected by humans on the mobile-device screen with the value of FU calculated by an algorithm from the digital image. This new colour comparator tool, renamed "Modern Forel-Ule scale", was produced commercially by an SME in Amsterdam, NL at the end of 2014 [http:// forel-ule-scale.com].

**Citizen water-colour monitoring apps.**   For a citizen to collect observations and use a smartphone's camera as a sensor to classify waters by their colour just by taking a picture of the water surface, Citclops designed and implemented an easy-to-use smartphone prototype

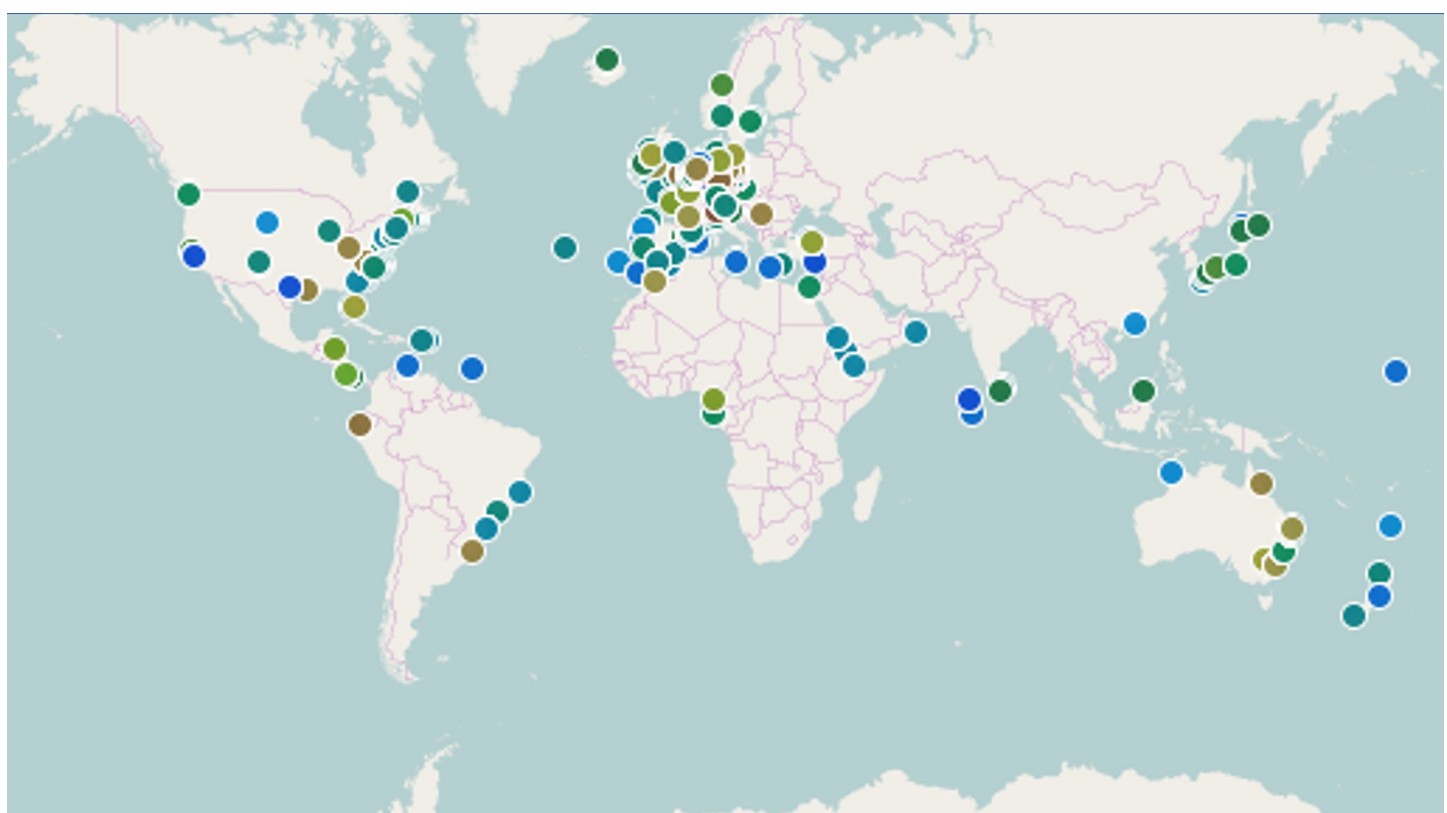

**Fig 8. Citclops's homepage shows all observations (about 9000, as October, 2019) collected through the EyeOnWater—Colour app and the Citclops—Citizen water colour monitoring app.** Images (photographs) are taken at sea by, for example, environmentally engaged captains, sailors and holidaymakers and inserted into to the database system.

app, called "Citclops—Citizen water colour monitoring" app, which has been developed and used by environmentally engaged people and the public in general to collect and understand scientific data. This smartphone app was built for two platforms, namely Android (versions 4.4–6.0) and iOS (versions 6.1–7.1), and became available in the corresponding app stores in summer 2014. Fig 12 shows different screens from the app. Key features are the comparison of the observed and photographed water colour with on-screen Forel-Ule coloured bars, and the final questionnaire to provide feedback on cloud conditions, precipitation and potential visibility of the bottom of the water body, all of them relevant for a quality assessment of the observed data. The "Citclops—Citizen water colour monitoring" app was declared obsolete in 2015, but can still be downloaded and used for testing purposes [DOI: 10.5281/zenodo.3507935].

At the end of 2014 the "Citclops—Citizen water colour monitoring" app was redesigned to make it more attractive (Fig 13, and new logo in Fig 14) and user-friendly and renamed "Eye-OnWater-colour" app (abbreviated as "EOW-colour" app). Specifically, the objectives of this redesign were:

- to develop an easy-to-use tool to collect and validate water-quality data, for the general public and for scientists;

- to create a modern solution to engage the public in environmentally relevant issues;

- to help oceanographers and limnologists to better understand the aquatic environment;

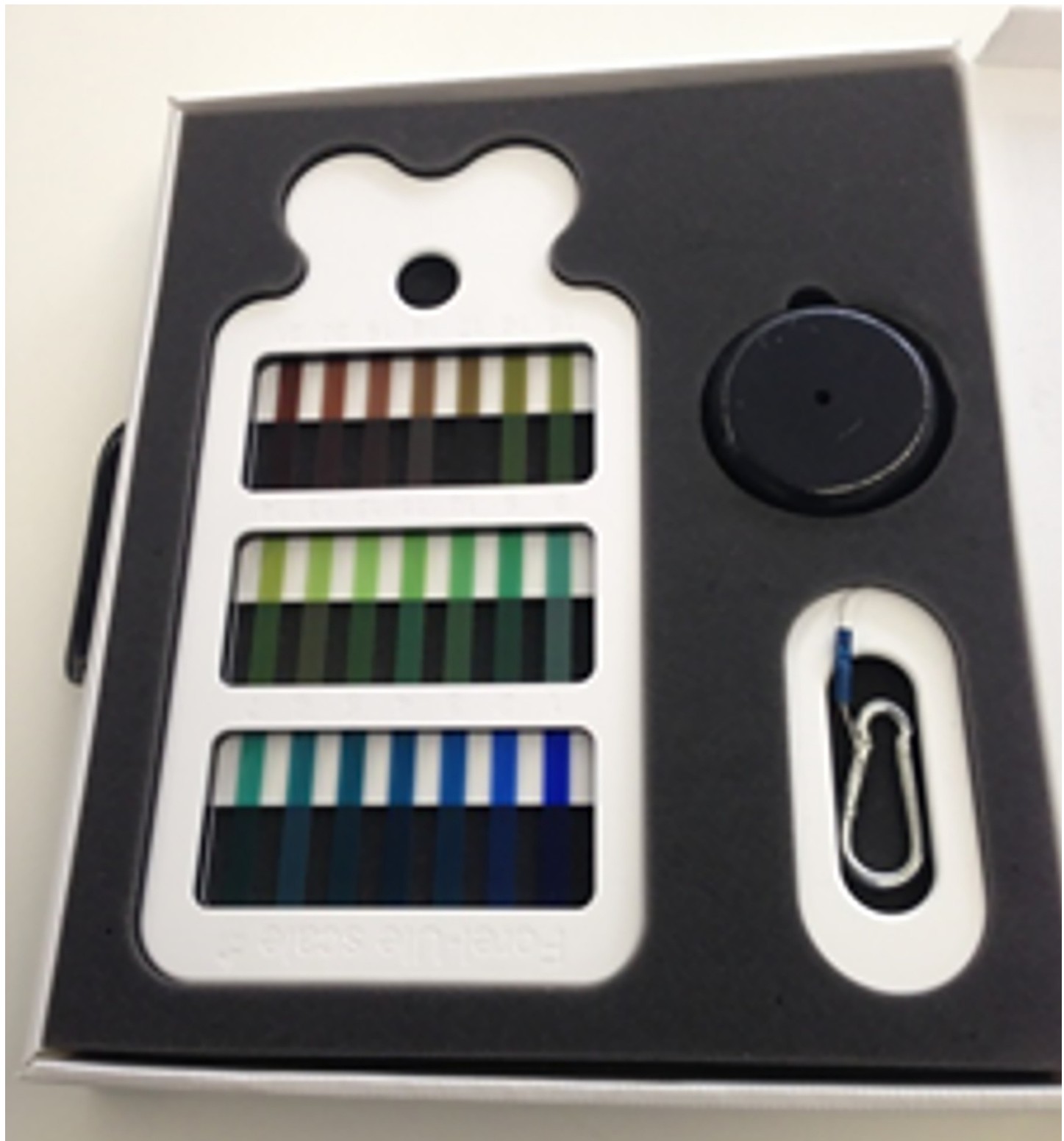

**Fig 9. The "Modern Forel-Ule scale" using plastic colour filters, together with a Secchi disk.**

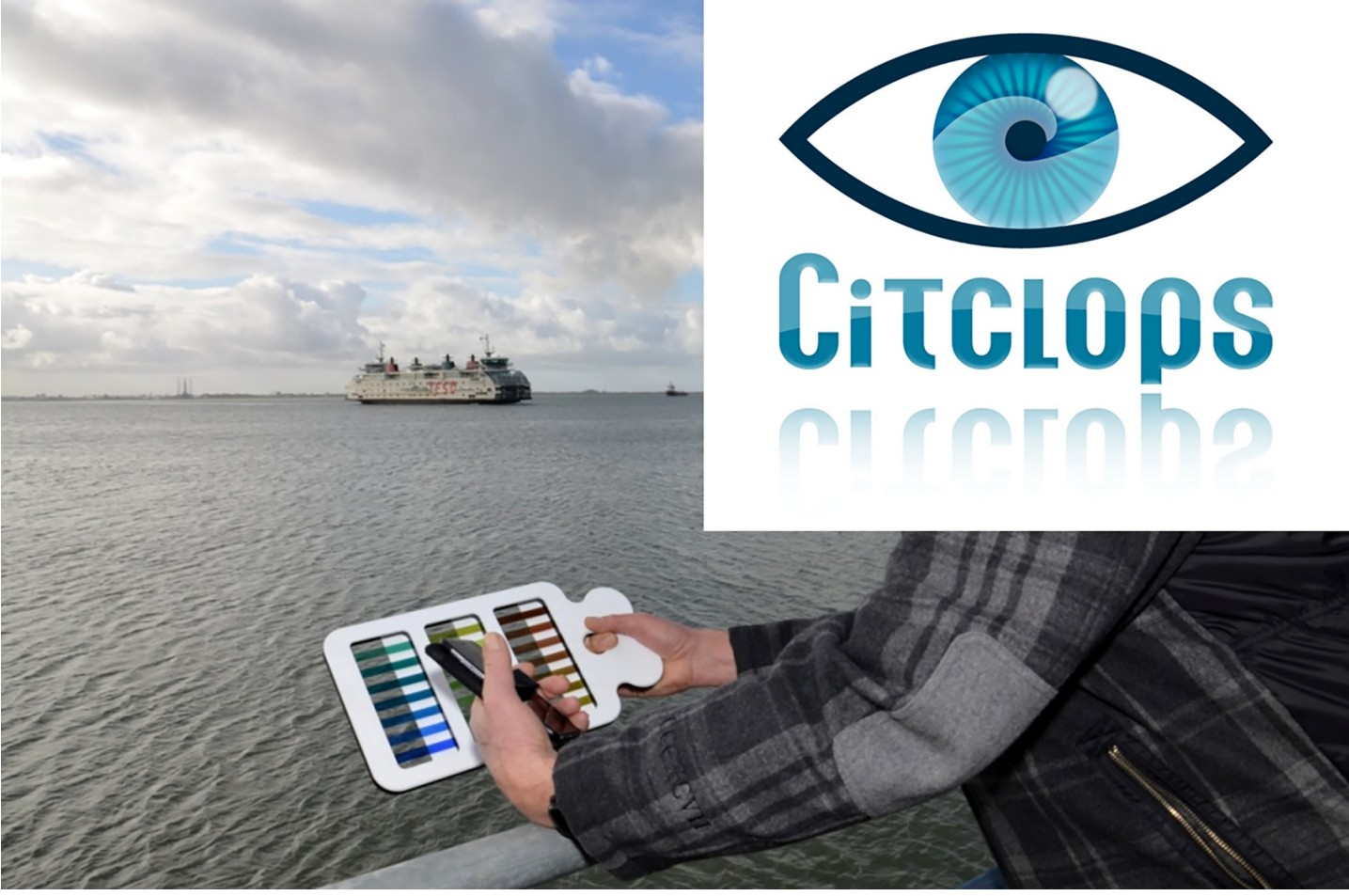

**Fig 10. The "Modern Forel-Ule scale" using plastic colour filters.**

- to contribute to quantitative research to map global-change impacts on the aquatic environment.

Part of the impact of Citclops on science is based on the use of the "EyeOnWater-colour" app to collect water-quality data. This data collection can be interrupted for a number of reasons, the main two ones are presented hereafter together with the corresponding solution. For the cases in which there is an interruption in data uploading from smartphone to server due to connectivity reasons (for example because there is no mobile coverage where the observations are collected, such as in open ocean), there is a caching mechanism in place to ensure uploading of data stored locally on the client (smartphone) to the data server as soon as the phone is online. For the cases in which there is an interruption in data recording by citizens, quantitative matchups and trend analyses of citizen data as well as earth-observation data are predominantly carried out on aggregated datasets. Hence the focus is on larger-scale features in time and space than at the scale of the individual pixel or citizen recording. This upscaling reduces the impact of outliers and irregular gaps in time and space.

Finally, logos for other water quality parameters have been designed for potential future apps to be used within the *EyeOnWater* (EOW) concept, as explained in the following section.

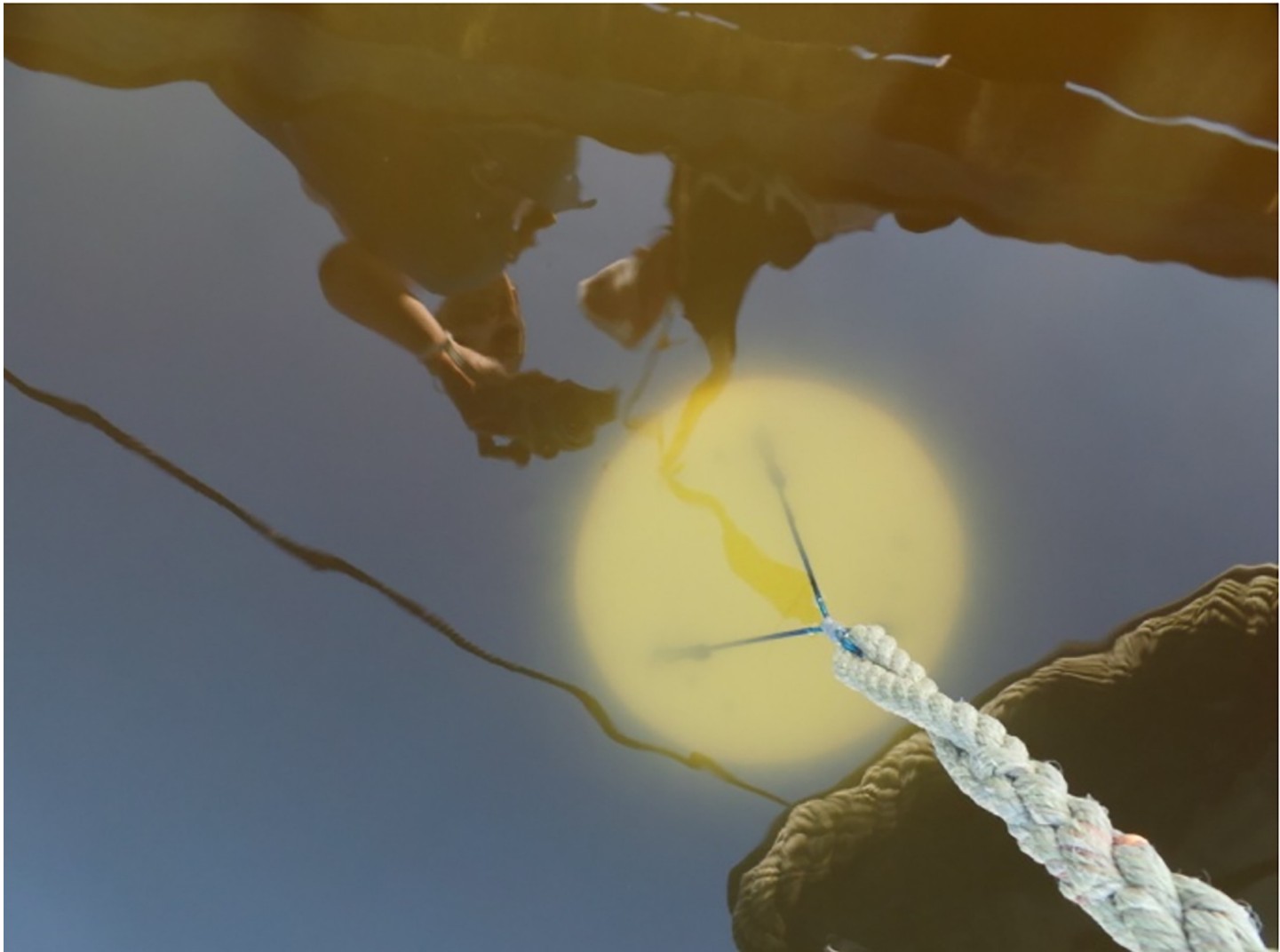

**Fig 11. A submersed Secchi disk.**

**The EyeOnWater website and data portal.** A new website [www.eyeonwater.org/] and a data portal have been created in October 2015 for the collection of water-quality data by citizens. This infrastructure allows the management of the collection of data not only on the colour of water, but also on algae, water clarity, fluorescence, macro-plastics, jellyfish and shells. Fig 15 shows the old and new logo for the website. Fig 16 illustrates the EOW data portal, hosting all data collected during and after the execution of Citclops together with a dataset of 300.000 historical observations.

Much effort has been spent on technical development of a new app and a new concept of how the app data can be used by experts, governments and how to make the link with citizens. An EyeOnWater domain has been claimed to facilitate the EyeOnWater concept to expand later to other parameters, e.g. fluorescence and KdUINO data, but also other ones.

Highlights of *EyeOnWater—Colour* are:

- The app has been developed to optimize the use on multiple devices and adapt to new iOS and Android version more easily.

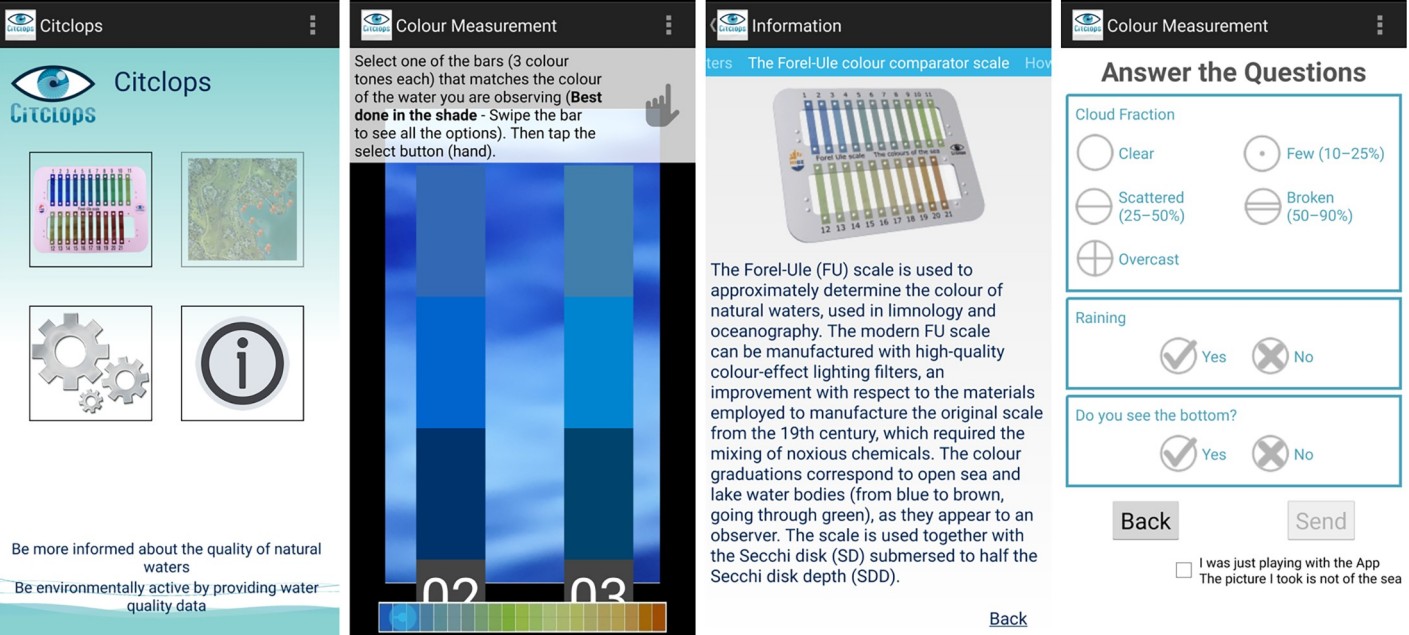

**Fig 12. The "Citclops—Citizen water colour monitoring" app.** From left to right: start screen, comparison of image colour and FU on-screen coloured bars, explanation page, and questionnaire page.

- The app can be found in the iOS and Android stores:

  - Google Play: [https://play.google.com/store/apps/details?id=nl.maris.citclops.crosswalk]

  - iTunes: [https://itunes.apple.com/us/app/eyeonwater-colour/id1021542366]

- The website [www.eyeonwater.org] provides app users a personal experience and shows all observations to all interested citizens.

The EyeOnWater website and adjacent "EOW-colour" app help users to assess the colour of natural waters. Water quality of natural waters can be estimated via their colour and transparency, since these optical properties are affected by the substances that are either suspended or dissolved in it. The "EOW-colour" app facilitates the monitoring of colour changes of natural waters. The app is based on an old oceanographic colour standard, the FU scale. The colour of the water body is compared with 21 on-screen colour bars.

**Education and contributor participation.** Several citizen-science experiments were carried out to collect data about water colour and help to understand the ecological status of surface marine waters in the open ocean. Besides citizens' activities and trainings in coastal zones, where the majority of water colour measurements were conducted [18], a number of trials were dedicated to areas not covered by the *CoastColour* remote-sensing project [http://www.coastcolour.org] (**Fig 17**). The Citclops/EyeOnWater app and static cameras were used by the crew to collect measurements of colour of surface marine waters during:

1. test navigations for the preparation of the ***Barcelona World Race*** (BWR) 2014/15 sailing regatta (**Fig 17**), carried out by the *GAES* team of the *Foundation for Ocean Sailing Barcelona* (FNOB);

2. the **Mediterranean Tall Ships Regatta** 2014;

3. the **BWR** 2014–2015 sailing regatta;

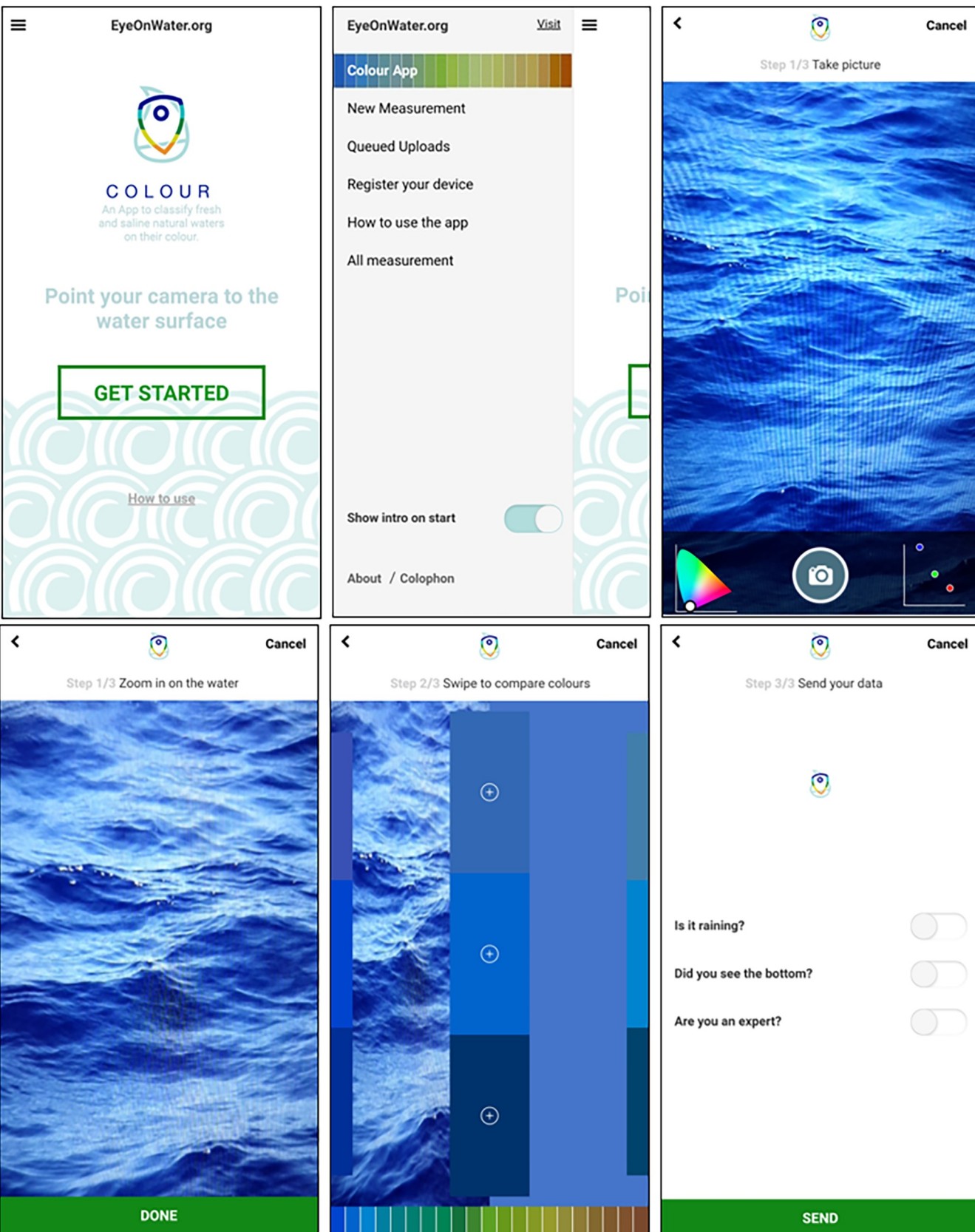

**Fig 13. Screenshots from the "EyeOnWater-colour" app: the initial screens; the chromaticity triangle and RGB spectrum; the FU-colour selection bars; and the location of the measurement on a map.**

4. the **Vendée Globe** 2016 sailing regatta.

The BWR in particular, the only double-handed (two crew per boat), non-stop, round-the-world regatta, in which all the skippers have been collaborating as scientific observers, was the opportunity to disseminate the results of Citclops through the event communication channels, because of the possibility to bring knowledge to a wider audience and to make issues related to the health of the oceans known worldwide. To introduce the Citclops project within this event there have been several preparatory actions. This process involved all of the boats taking part in the BWR 2014/15. In this edition of the race there were eight teams formed by two skippers (Fig 18). It was decided to distribute a smartphone with the Citclops app installed to each team. There have been also some introductory and test sessions with the race teams. During these sessions the operation and purpose of the Citclops app has been explained in detail, and a demonstration of its operation and a test of data-sending via satellite have also been performed.

To make the process of adaptation to the Citclops app easier and to avoid doubts, a small pictorial manual has been prepared, showing the basic steps to take samples and how to send them via computer and the Internet (**Fig 19**). Offshore, there is no wireless connection accessible by the smartphones used; for this reason, the samples are copied to the computer located on the boat and then, to be able to track them in near real time, sent by email. All boats have wireless connection to satellite and connect to the Internet several times a week.

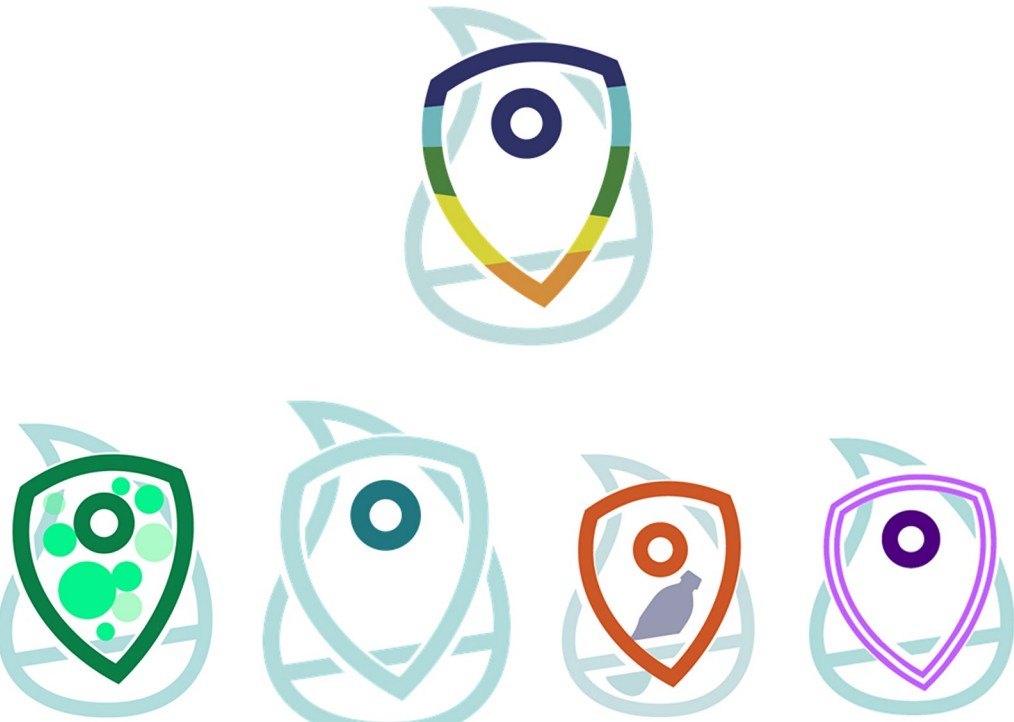

**Fig 14. The EyeOnWater logos.** Top: EOW-colour. Bottom: examples for future apps.

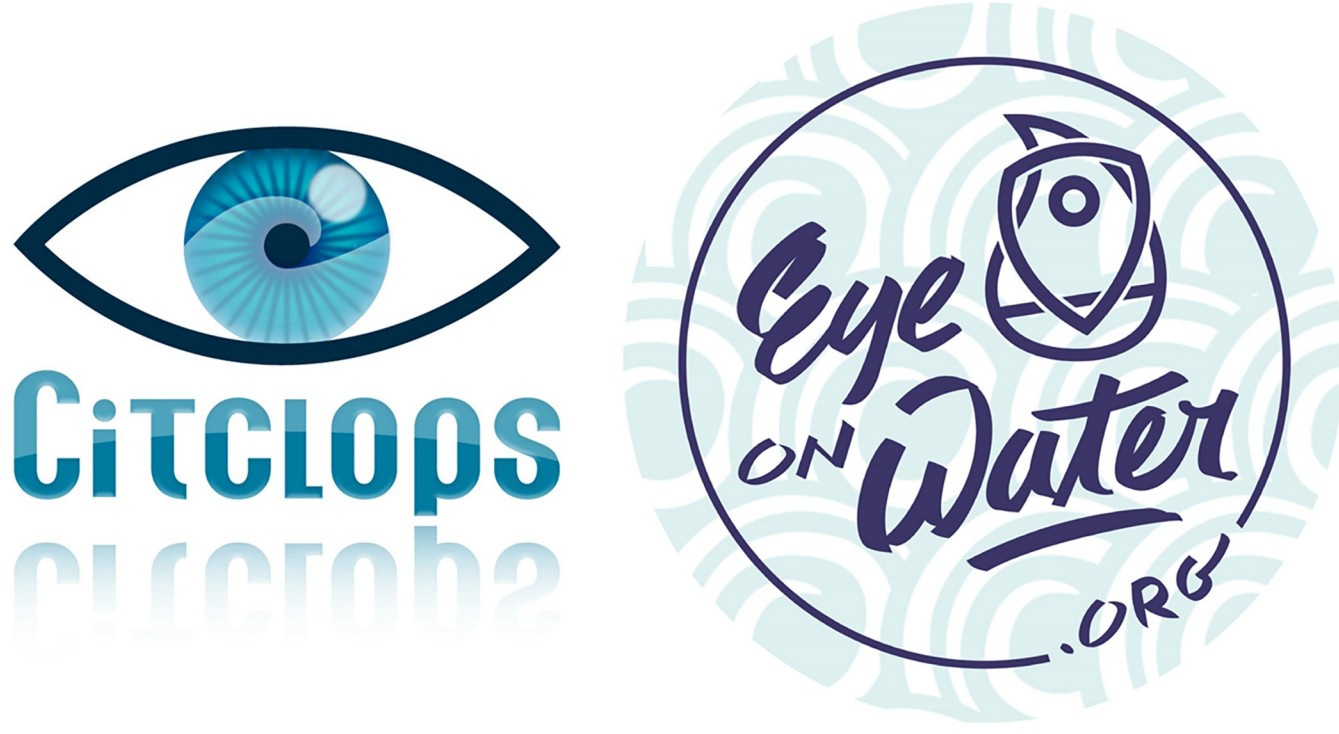

**Fig 15. The old and new logo for the website dedicated to the collection of water-quality data by citizens.**

In addition to the printed manual, an instructional video was released showing how to use the app step by step. This video was stored on the smartphone to be viewed when necessary. A

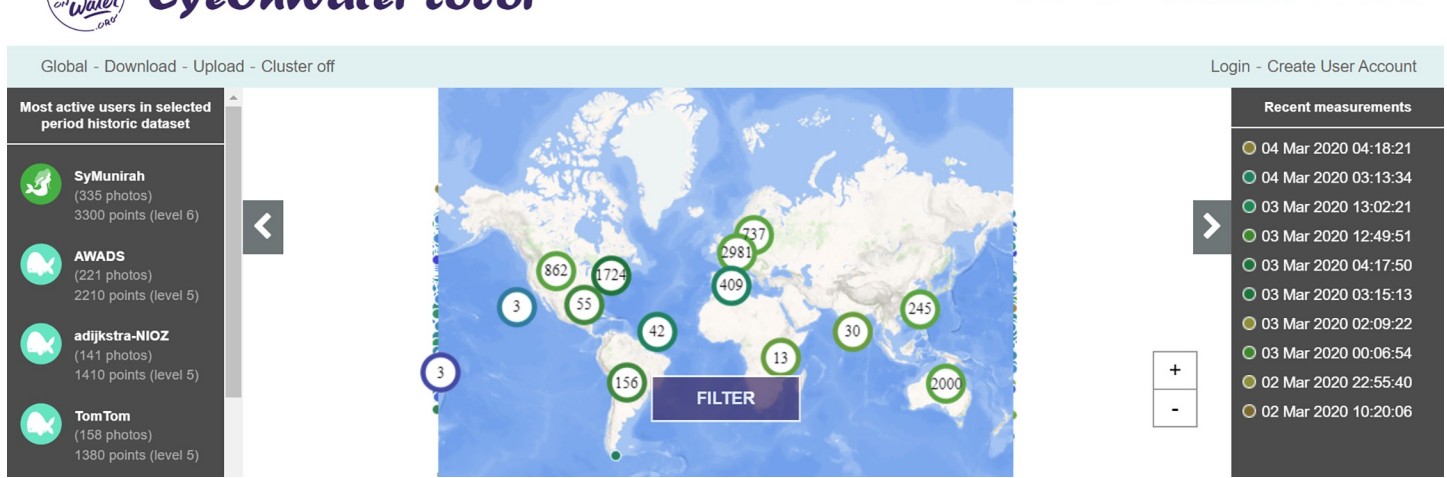

**Fig 16. The EyeOnWater website, where the user can navigate through all the observation samples of water taken by users of the Citclops smartphone application or uploaded manually using the upload web-interface (accessed on October 2019).**

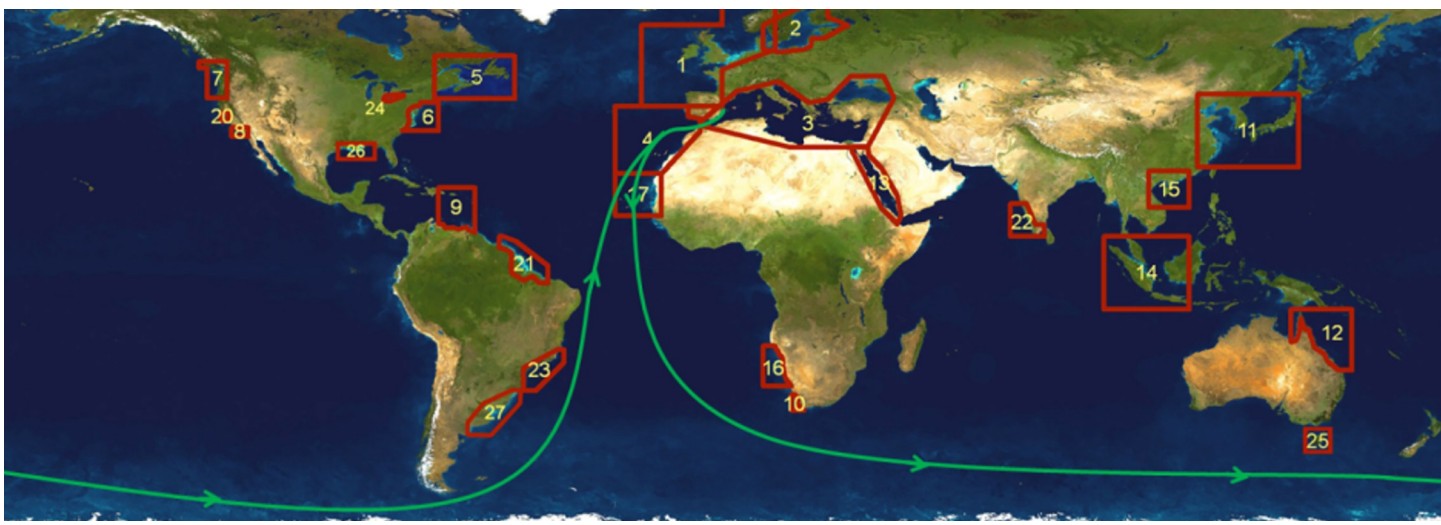

**Fig 17. CoastColour project's regions and *BWR 2014/15*'s course (source [http://www.barcelonaworldrace.org]).**

**Fig 18. Barcelona World Race's start on 31/12/2014.**

few days before the race, in a plenary meeting with all teams, the Citclops project and the app were officially presented to the media. To track participation during the course of the race a dedicated web page was implemented [http://citclops-barcelonaworldrace.weebly.com/]. This website also informs the citizens about this experiment and promotes better environment management. After completion of the race the website remained online and provides access to these results openly.

## Water transparency

Two systems have been developed to determine the water transparency:

1. a "do-it-yourself", low-cost, moored system (KdUINO) to measure the diffuse attenuation coefficient [20]; and

2. a system (TrandiCam) which involves the use of a set of plastic targets that can be photographed undersea. (This system is not documented in this paper and more information can be found in [18].)

**KdUINO—Measuring the diffuse attenuation coefficient.**  The KdUINO is a low-cost, moored system (a buoy) that measures the diffuse attenuation coefficient ($K_d$). $K_d$ relates the attenuation of light to the properties of the water through which the light is travelling and to water transparency [19], which is not a physical parameter. The KdUINO, described by Bardaji et al. [20], is based on an open-source hardware platform that controls and stores the data obtained from several light sensors placed at different depths. Post-processing analysis allows retrieving $K_d$ from the values obtained by the sensors. Also, the instrument design minimizes measurement errors due to sunlight fluctuations, allowing reliable measurements in the first meters of the water column. Due to this key feature, the KdUINO can be used in areas with shallow waters. With an Android app, the "KdUINO Remote Control" app, users can control and receive the measurements of the KdUINO. The "KdUINO Remote Control" app (Fig 20) allows contributors to receive and share (via e-mail) the measurements of the buoy. With the Python code provided in the supplementary materials by Bardaji et al. [20], researchers and volunteers can analyse the KdUINO data.

Similarly to the colour map, an interactive map to see all KdUINO data has been developed. In this case, the authors have used the CARTO open-source platform [carto.com] to make the maps. The authors have developed several versions of the map, oriented to different types of users of the data. One of the maps is shown in Fig 21 and can be found at [https://bit.ly/kdumap]. The authors used the Python API of CARTO [https://github.com/CartoDB/carto-python] to upload the KdUINO data automatically as they are received. Measurement dots are painted with colours representing the value of $K_d$. On the map, the colour of the points is from lighter, for clear water or low $K_d$ values, to darker for muddy water or high $K_d$ values. Users can quickly view where the water is more transparent.

**Education and contributor participation.**  In relation to KdUINO, a collaboration was established with Plàncton, Divulgació i Serveis Marins, an enterprise offering dissemination, scientific outreach and marine-related services: they have received the necessary material and the training to be able to use the KdUINO buoys with their customers. The observation of transparency was integrated in their "scientific snorkelling" (guiding the customers in the sea and explaining what they see); they also explained how to build a KdUINO. Another part of their work was related to education: KdUINO buoys were used by schoolchildren to collect measurements. Also, the authors worked with several secondary schools, organizing

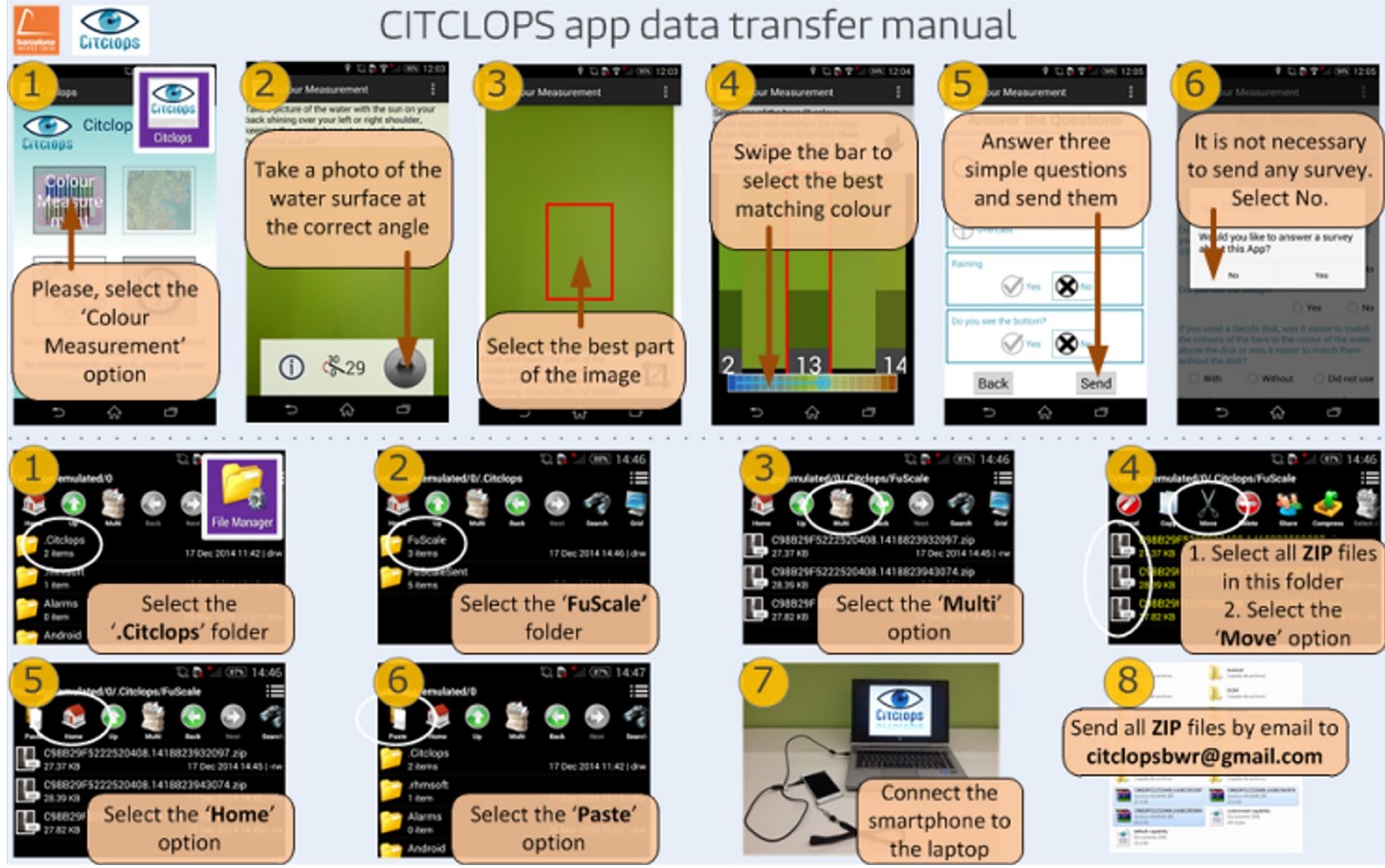

**Fig 19. Printed manual delivered to BWR skippers.**

workshops to help the pupils to develop their own KdUINOs. The workshops have been presented several times in different media canals as showed in **Fig 22**.

Also, the authors worked with regular divers, who have committed to taking pictures for the project when they dive, and with professionals working on boats.

## Water fluorescence

**SmartFluo: A smartphone adapter for fluorescence measurements.** Within Citclops, a prototype sensor system for fluorescence measurements with smartphones has been constructed (SmartFluo) (**Fig 23**) [28–33]. This allows fluorescence measurements under controlled and standardised conditions. A 3D-printed housing, attachable to a smartphone, ensures that all measurements are conducted in dark conditions, so the signals are not disturbed by ambient light. An external high-power blue LED (Lumileds Holding B.V.'s LUX-EON Rebel) is used to excite fluorescence in the water sample in a cuvette. Measurements are automatically conducted by means of a smartphone application. The external LED is triggered by means of a sound signal via the audio phone connector. The red fluorescence signal then passes a low-cost red filter and is recorded by the internal smartphone camera, resulting in an image in jpg format, consisting of *red-green-blue* (RGB) pixels. The centre of this image is cropped and analysed.

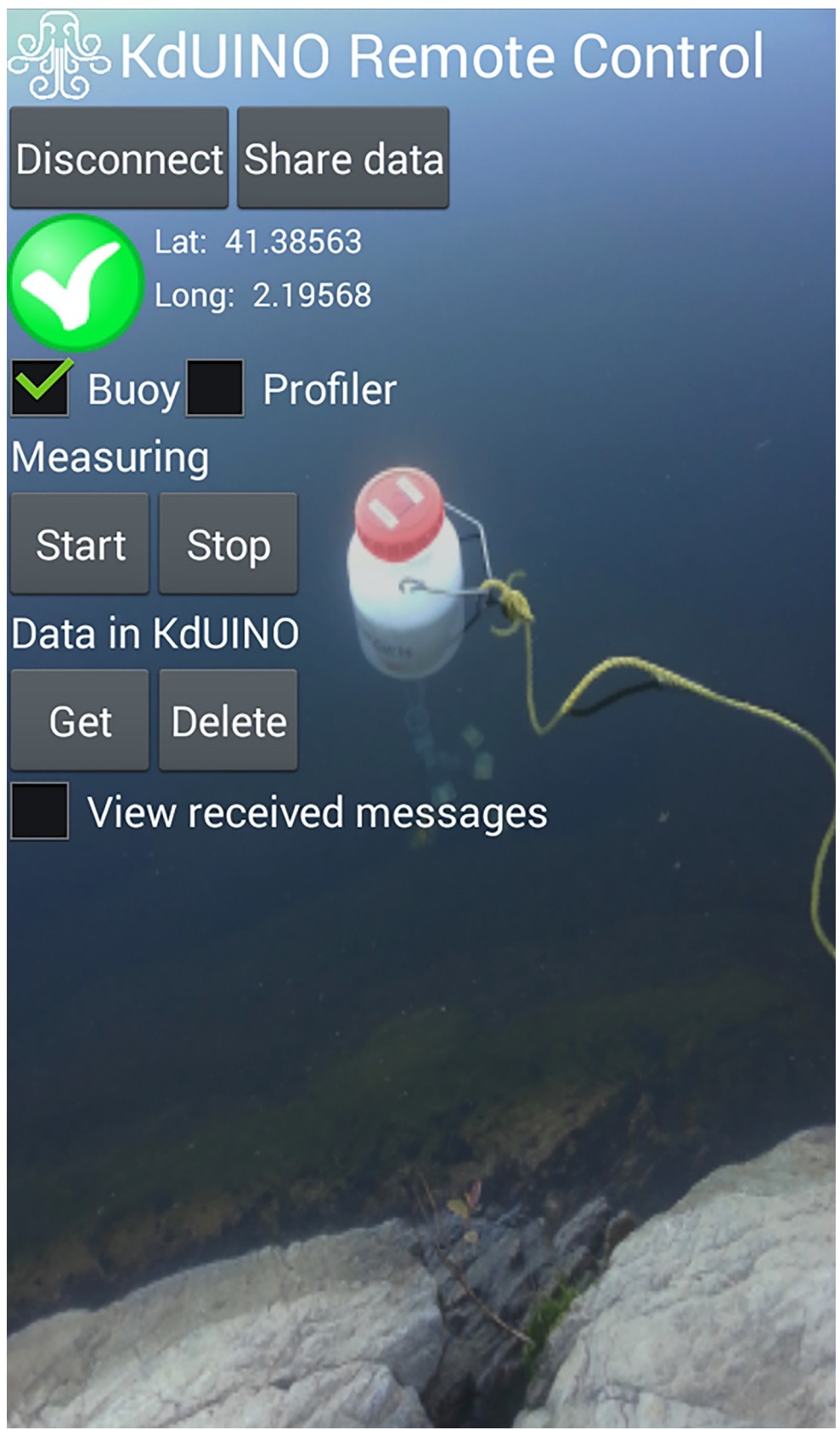

**Fig 20. Screenshot of the "KdUINO Remote Control" app.**

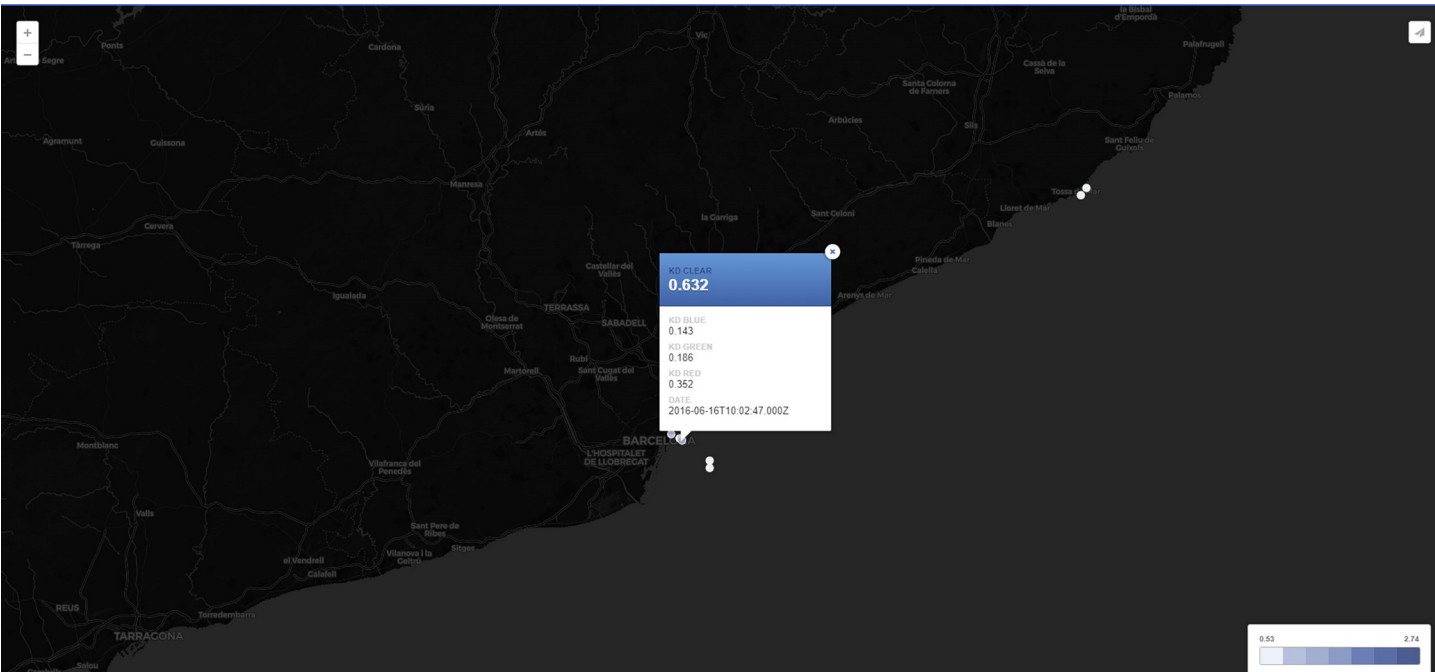

**Fig 21. Visualisation of the KdUINO data on a webpage.**

A novel algorithm to calculate the Chl-*a* fluorescence signal from the RGB image has been defined (RGB2Chl). The algorithm performance was evaluated under controlled conditions, with a dilution series of Chl-a extracted in acetone and at room temperature. A high linear correlation between the intensity of the red channel ($I_{red}$) and Chl-a concentration was found in the lower Chl-a concentration (1–30 μg L$^{-1}$) with $R^2$ = 0.98 (**Fig 24A**; continuous line). A regression over all data found an exponential fit with $R^2$ = 0.99 (**Fig 24A**, dotted curve) [28].

Control measurements conducted with a *PerkinElmer LS 55 Fluorescence* bench-top fluorometer (LS 55) confirm the applicability of the SmartFluo with the RGB2Chl algorithm as a new method for measuring Chl-a fluorescence. A linear regression is applied between intensity of *in vitro* fluorescence maximum of LS 55 at 670 nm and $I_{red}$, as recommended by Arar and Collins [34] for quality control (**Fig 24B**), at low Chl-a concentrations (1 μg L$^{-1}$ to 30 μg L$^{-1}$). An $R^2$ of 0.99 (**Fig 24B**, continuous line) confirms that $I_{red}$ is properly recorded and determined by the SmartFluo with the RGB2Chl algorithm, with the assumption of linear behaviour. An additional polynomial fit (**Fig 24B**, dotted curve) represents the sensibility curve of SmartFluo with the RGB2Chl algorithm ($R^2$ = 0.99) [28].

**MatrixFlu: A compact multi-wavelength fluorometer.**    Within Citclops, a multi excitation sensory system was constructed. This system has the capability of measuring multiple parameters with an intelligent selection of four different LEDs for excitation and photodiodes with four different filters for detection of fluorescence. The principle of multi-parameter sensing was implemented into this new portable sensory system, the MatrixFlu, representing a "quasi"-excitation-emission-matrix [26]. The optical window of MatrixFlu is made of quartz silica with nano-coating to avoid biofouling. For this instrument, LEDs, filters and detectors for different wavelengths were selected and tested; and several parameters can be measured simultaneously with the device (Table 1). The MatrixFlu has a built-in web server, so it can be controlled, via a Wi-Fi connection, by any smartphone or tablet, independently of the operating system. Signal transmission is realised by using fibre optic cables leading the light from

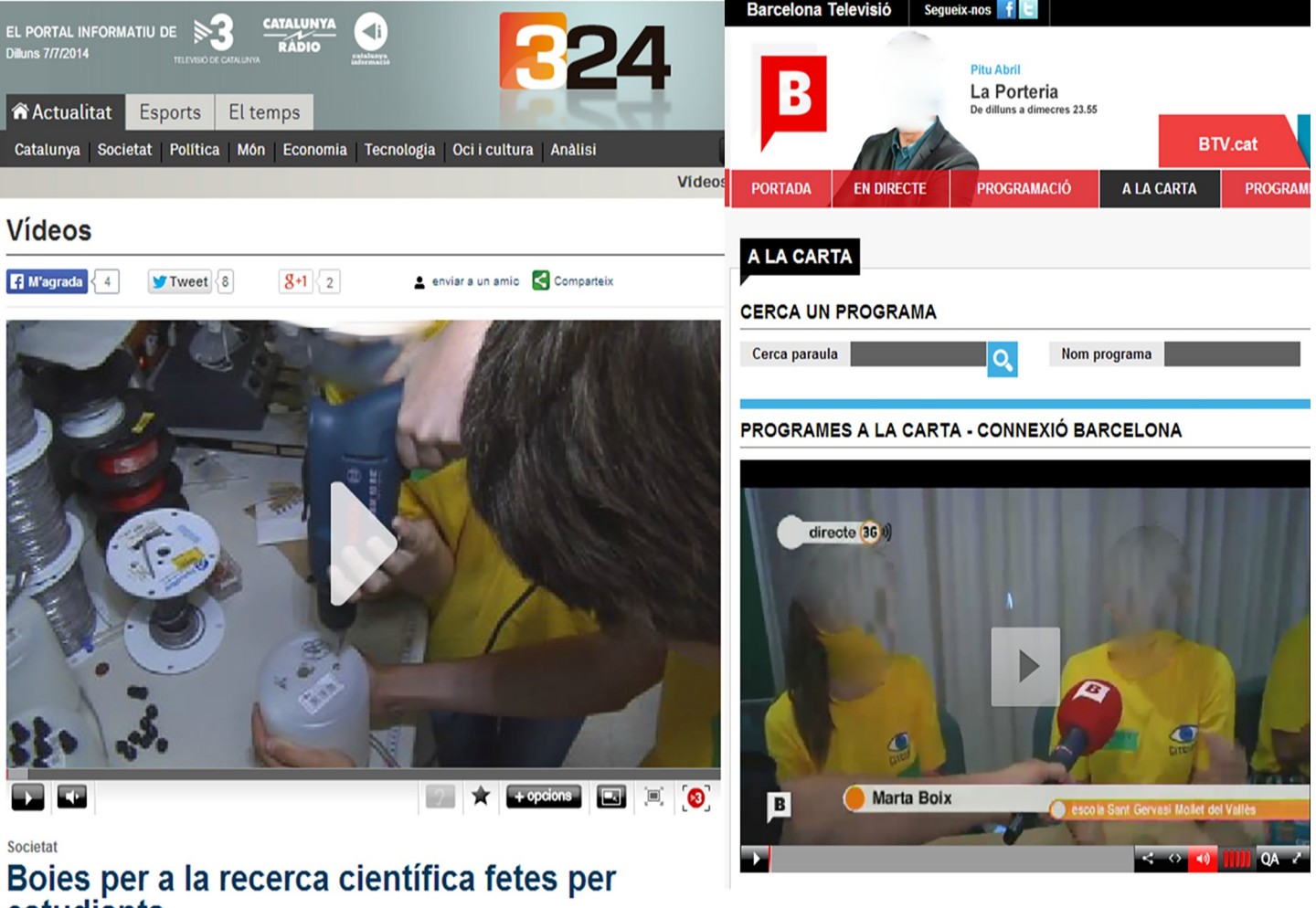

**Fig 22. KdUINO workshops presented by Spanish media.**

LEDs in the interior through the front-end detector board into the water. Due to this construction, a robust miniaturized housing of MatrixFlu is possible and is made of stainless steel with a diameter of 36 mm and a length of 155 mm (Fig 25).

In this way the hand-held portable device can measure several parameters in the water by means of fluorescence. Chl-a, fCDOM and phycocyanin are addressable fluorescing substances in water, as well as light scattering particles with high wavelength (Table 1). Depending on their chemical structure, the organic substances differ in their fluorescence emission and give thereby the opportunity to discriminate between several components/fractions of fCDOM, represented by fCDOM1, fCDOM1 and fCDOM3 (Table 1), as they have the same excitation wavelength while their emission differs [26]. The scatter detection at 850 nm (Table 1) is used to correct for turbidity.

**Education and contributor participation.** Application of both instruments (SmartFluo and MatrixFlu) is straightforward and they are easy to handle. The MatrixFlu was tested within another EU funded project: Next generation, Cost-effective, Compact, Multifunctional Web Enabled Ocean Sensor Systems Empowering Marine, Maritime and Fisheries Management (NeXOS, [26]). The SmartFluo was used under guided conditions (by researchers) during

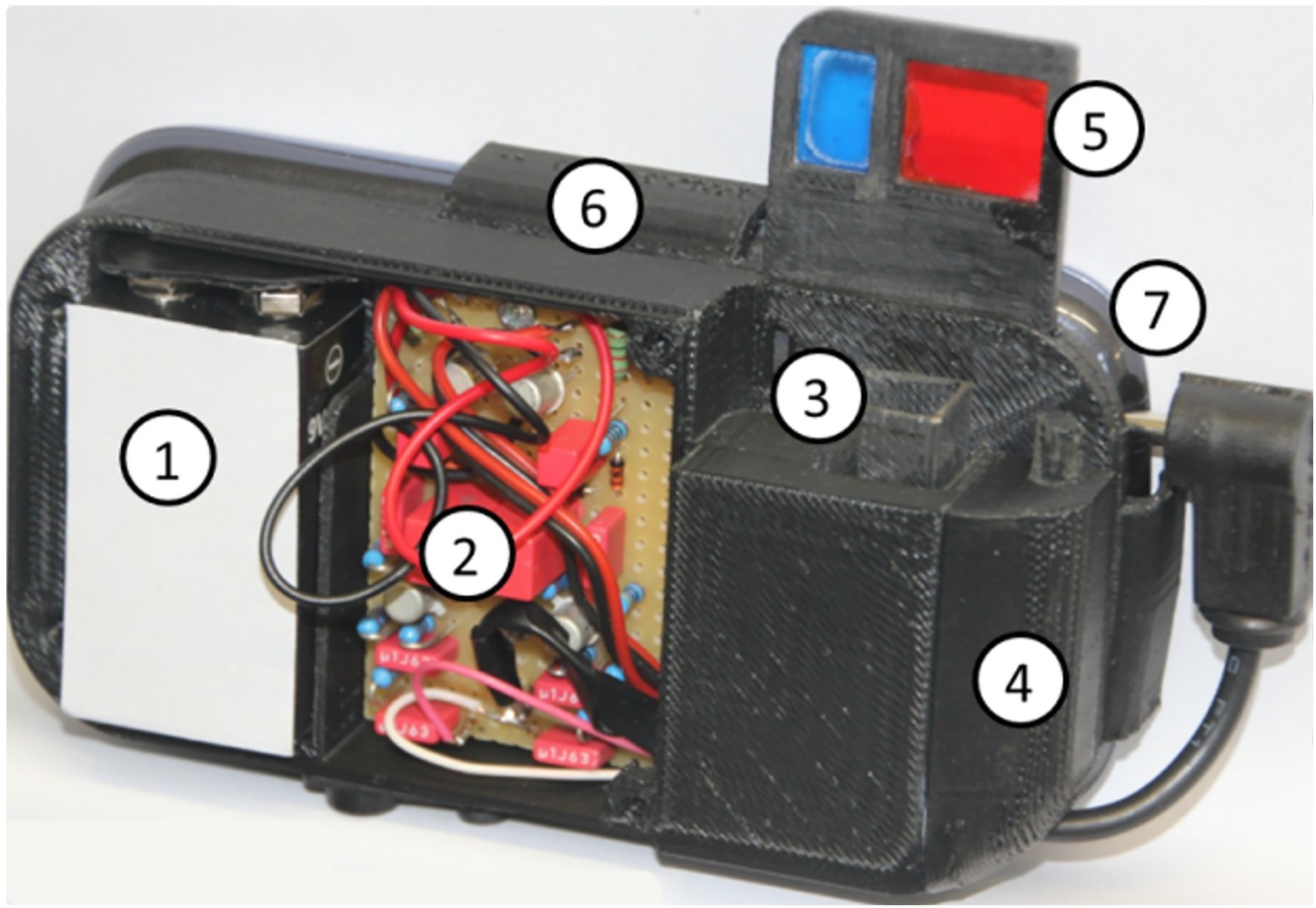

**Fig 23. Current setup and prototype of SmartFluo, including: 1) battery power supply, 2) circuit board, 3) cuvette, 4) covered LEDs, 5) filter setup, 6) holder system, and 7) smartphone.**

citizens' training events of the project, which include a sea kayak workshop in Llançà, Spain (March 31 –April 1, 2015), the Blue Info Days in Wexford, Ireland (April 24–26, 2015), the Ocean Sampling Day (June 21, 2015), and during scientific research campaigns [18].

### Delivery of information to citizens, decision makers and researchers

Typical activities included in citizen science range from data collection, quality-control and analysis to information access and delivery. It is then important to define *data*, *information*, and the related concept of *knowledge*, and their relationships to one another. Understanding their differences is a key to increasing knowledge and competence. Data are collected and analyzed to create information suitable for making decisions, while knowledge is derived from extensive amounts of experience dealing with information on a subject.

One of the main objectives in EyeOnWater/Citclops is to deliver information to citizens, decision makers and researchers. This information is provided through tools capable of retrieving, quality-control, analysing, interpreting and visualising marine data of interest. These tools have personalised interfaces that depend on the users' requirements. To achieve their development, methods involving advanced data analytics are used to interpret collected

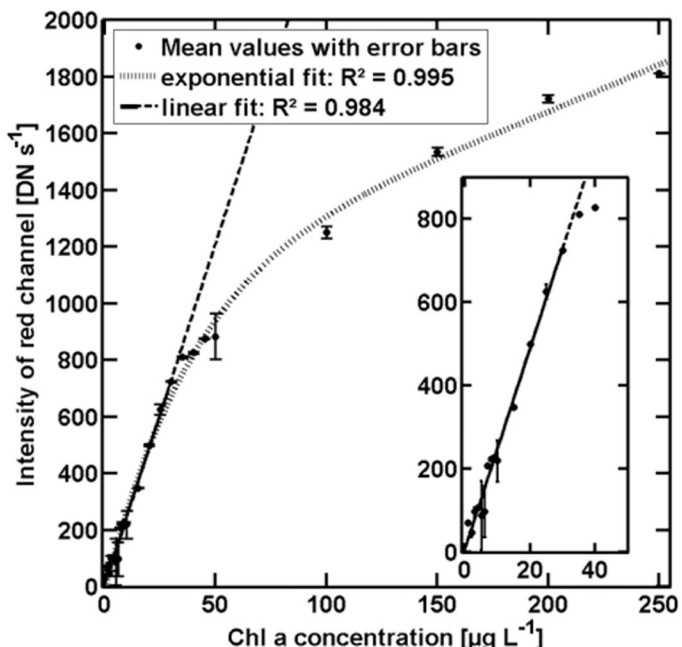
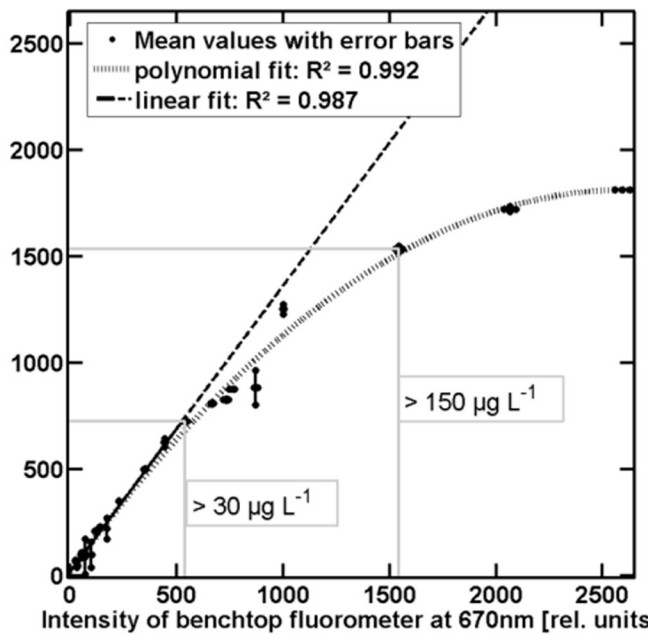

**Fig 24.** a) Intensity of the red channel of smartphone RGB-images as function of the Chl-a concentration including error bars derived from triple measurements (dots) with linear fit (black) for Chl-a concentration from 1 µg $L^{-1}$ to 30 µg $L^{-1}$ and exponential fit (dotted line) over the full dilution series. Dashed parts of fitting line depicted range not included into the fit. The inside figure highlights linear range at low Chl-a concentrations. b) Intensity of the red channel of smartphone RGB-images as function of the maximum intensity of the LS 55 at 670 nm. Triplicates (dots) with error bars are plotted, including linear fit ($R^2$ = 0.99; continuous and dashed line) in low Chl-a concentrations: 1 µg $L^{-1}$ to 30 µg $L^{-1}$ and polynomial fit (dotted) over full dilution series ($R^2$ = 0.99). Additional related Chl-a concentrations are marked in two ranges: > 30 µg $L^{-1}$ and > 150 µg $L^{-1}$ (right of the grey boxes).

data of water colour, transparency and fluorescence, and additional context variables of interest to the end-users. The purpose of this section is

- firstly, to describe the end-user applications, testing and user feedback of a *decision support system* (DSS), the Citclops Data Explorer, with special emphasis on the influence of the context and the user profile;

- secondly, to define the interfaces among the different modules developed in this research;

- thirdly, to evaluate the Citclops Data Explorer through demonstration scenarios.

This work focuses on the development and implementation process of the *Citclops Data Explorer*, a decision-support system to be utilised within the context of citizen science.

**End-user applications, testing and user evaluation.** A web-based data-visualisation application was developed, that displays the data collected by the citizens through the

**Table 1. Current setup of MatrixFlu in visible wavelength range where non-allocated fields are shaded.**

| Excitation [nm] | Detection [nm] | | | |
|---|---|---|---|---|
| | 460 | 655 | 682 | 850 |
| 375 | fCDOM1 | fCDOM2 | fCDOM3 | |
| 470 | Scatter 460 | | Chl-*a* | |
| 590 | | PC pigments | PC via Chl-a | |
| 850 | | | | Scatter |

PC = phycocyanin (blue pigment found in algae); fCDOM1, fCDOM1 and fCDOM3 address different fCDOM components.

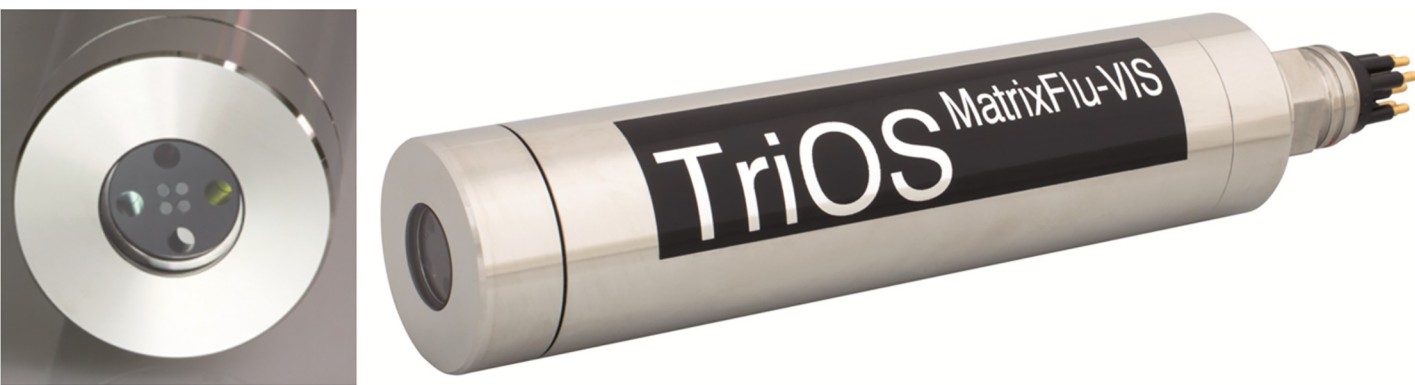

**Fig 25. Appearance of MatrixFlu (Courtesy: TriOS GmbH, Germany).**

smartphone applications, and other data supplied by different data sources (satellite colour, weather, and bathing water quality indicators) and presents information after data processing by the DSS algorithms.

Initially, the DSS interfaces had been defined as three top-level interfaces, corresponding to three types of foreseen users: *citizens* (or general public), *decision makers*, and *researchers*. However, after testing and evaluation through a focus group which included the three types of end-users, an important lesson learnt was that the user-based labels in the DSS interfaces generated a negative feedback from stakeholders and end-users; the labels were remarked as being exclusionary, particularly to citizens who may have had an interest in seeing other (non citizen-oriented) interfaces. In fact, all users could access all interfaces at any time, but this is not what users perceived. After considering this feedback, a decision was made to re-brand the DSS interfaces on the basis of their functionality rather than their target end-user.

Thus, the top-level label for the set of DSS interfaces has been defined as the "Citclops Data Explorer" (Fig 26) comprising of three views, accessible by any user category:

- *Citizen Observations* (EyeOnWater) (Fig 16) [http://www.eyeonwater.org/]. Through this interface, the user can navigate through all the observation samples of water taken by citizen contributors using the EyeOnWater/Citclops smartphone applications or uploaded manually using an online interface provided. Furthermore, the user can assist in the quality control of the data by "flagging" measurements and samples which they deem to be inconsistent.

- *Marine Data Analyser* [http://citclops-data-explorer.herokuapp.com/marine-data-analyser]. Through this interface (DOI: 10.5281/zenodo.3668704), the user can observe more in-depth information on the state of seawaters such as the historical state of bathing waters or seawater colour obtained from satellite observations. These data are collected from different sources and analysed by machine-learning algorithms to predict future trends (predictive analytics). The results are shown by graphs or specific marks on the map with colour codes indicating risks (e.g. bathing water contamination).

- *Marine Data Repository* (Fig 27) [http://www.citclops.eu/search/welcome.php]. Through this interface, users are able to easily download all Citclops data (from citizens, satellites, jetties, sensors) and work with their own analysis tools for their own purpose (e.g. study).

In this way, the tools to consult all data and information gathered and produced during the project are more accessible and understandable for any type of user, whether they are citizen

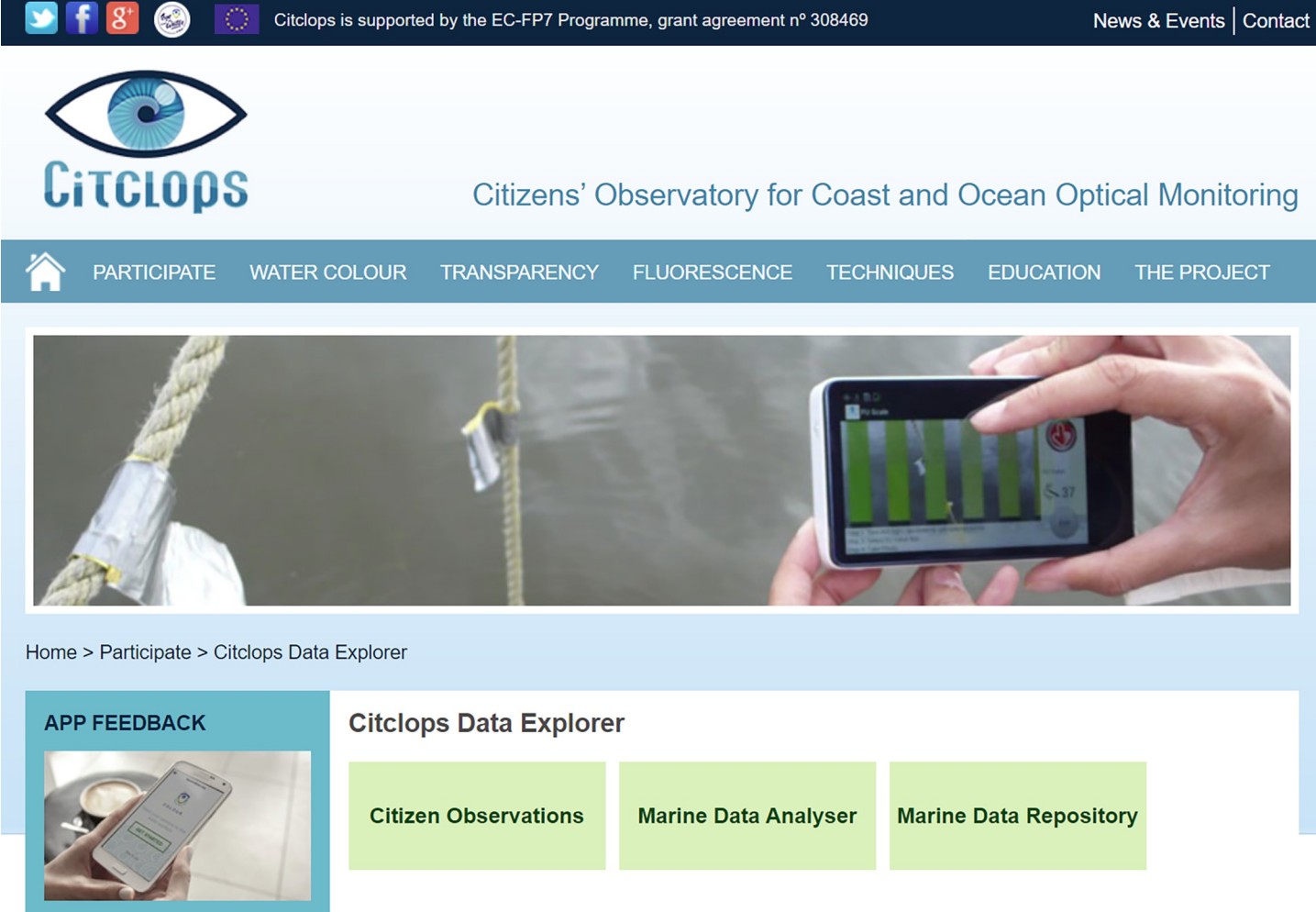

**Fig 26. "Citclops Data Explorer" access interface.**

users, decision makers, researchers, other users or they have overlapping profiles. Through the re-branding the intention was to invite the user to discover new possibilities and features of the applications. Thus, access to extra information does not discriminate against any user type. The citizen user feels now more comfortable during the navigation process and is not discouraged from exploring the more advanced functionalities of the Marine Data Analyser and Marine Data Repository.

The *Citclops Data Explorer* interfaces present several advanced features to improve functioning and usability, such as:

1. a clustering system with an upper limit, so that users do not end up with a single cluster if they zoom out very much, but still see a spatial distribution of where attributes have been measured;

2. the display of a time series when a right mouse click is performed on a cluster representing a series of measurements in the same exact location (Full pin-overlap is allowed only when different measurements are done at the same exact location, for example in the case of a moored instrument.);

3. the colouring of each cluster marker using the underlying values, e.g. mean FU colour;

4. the possibility to define a free date-range for data search and download;

5. the possibility to select a date-range for data visualisation;

6. bathymetry and water-colour from satellite provided as layers, which are explicitly activated by the citizens in certain visualisations, so they always know what they are being presented with;

7. adaptive interface made friendly to smart phones and tablets, including fast map-loading;

8. centre of view personalised according to the profile or location of the user, if available; if not available, the Mediterranean area is used as the default centre;

9. possibility to visualise metadata about citizen samples on the map-based interfaces through short and extended metadata pop-ups (Fig 28 and **Fig 6**);

10. possibility to flag inconsistent or irrelevant samples in the dataset thus allowing the citizen to contribute to data quality control.

## Beyond Citclops

The continuity and sustainability of the functionality of the Citclops Data Explorer beyond Citclops have been studied in depth, and in this section a comprehensive analysis in this sense is presented for the full list of Citclops components (and sub-components) that the *Citclops infrastructure* consists of, considering the dependencies among components and the sub-infrastructure needed for each component. (Note that components do not need to all be on one machine but can be distributed over different machines.) This analysis includes, at the end, recommendations that can be useful to any citizen-science project, about the approach to follow in case of needing to migrate the project services from a short-term hosting system to a long-term one.

### Full list of Citclops components

**I. Citclops infrastructure**

a. Operating system: Debian Linux 6.0.8 (Squeeze)
All Citclops components run on Debian Linux 6.0.8. The machine with the operating system should either be a dedicated server or a virtual machine that is online.
Dependency: None

b. Database server: PostgreSQL 8.3 with PostGIS extension 2.0
Dependency: I.a

c. Geoserver 2.4.4 running on Apache Tomcat 7.0.2
Dependency: I.a

d. Python 2.7.9 with libraries listed in S1 Appendix
Dependency: I.a

e. Octave 3.8.2
Dependency: I.a

f. Apache 2.4 with mod_wsgi enabled
Dependency: I.a

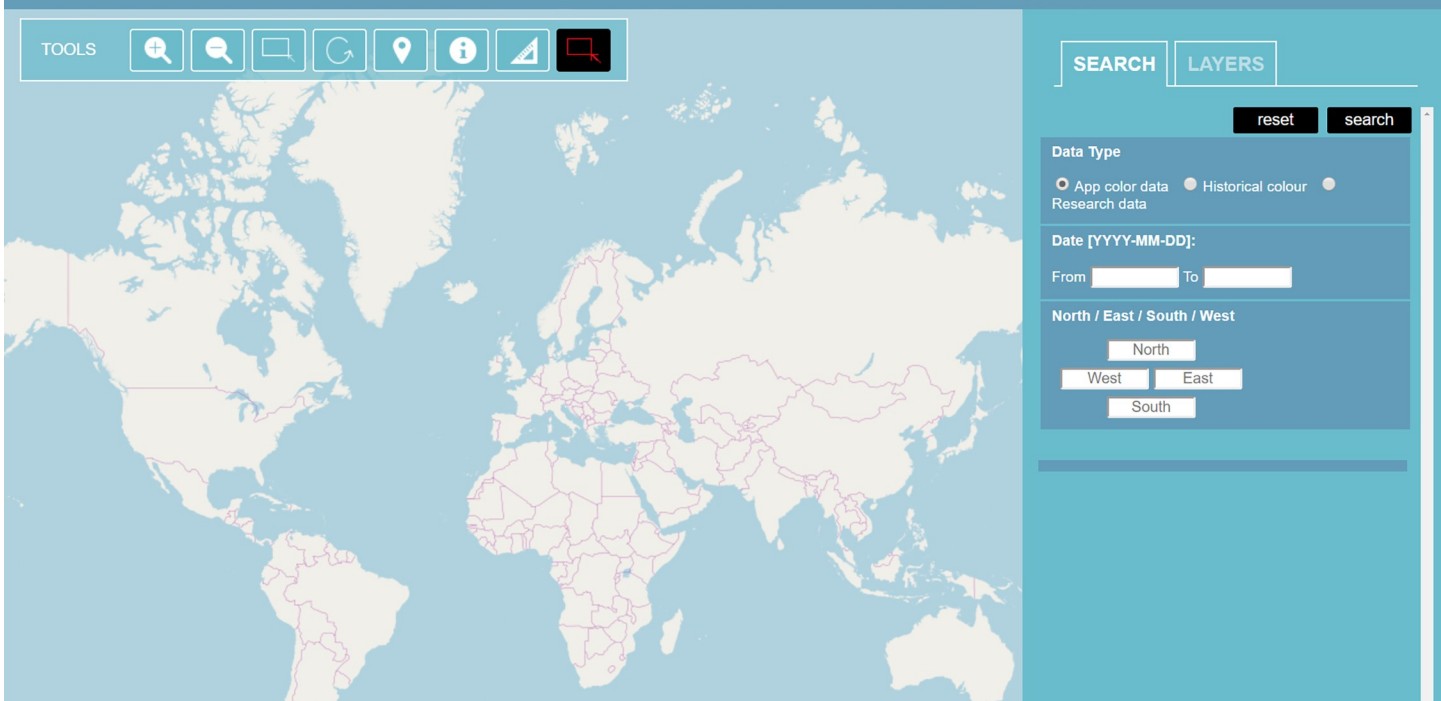

**Fig 27. "*Marine Data Repository*" interface.**

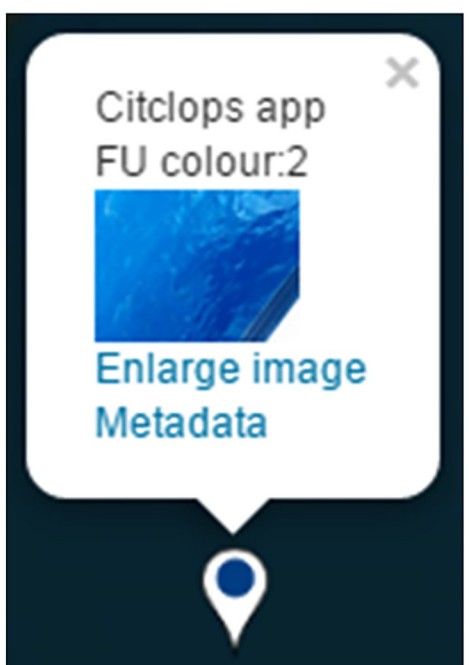 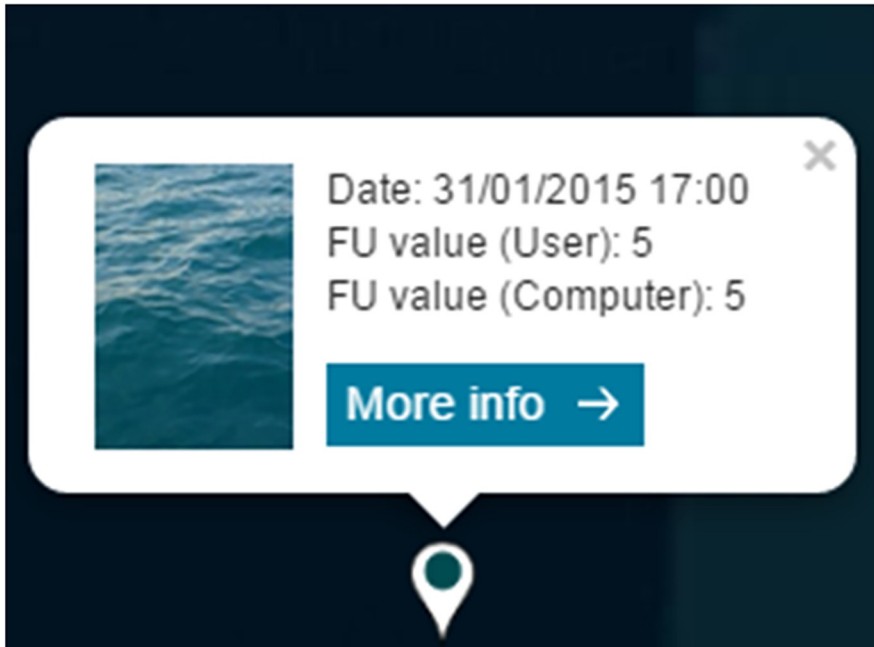

**Fig 28. Short metadata pop-up.**

**S. Supporting services**

a. Upload service: This service/module receives observations/images and meta-data XML from the Citclops iPhone and Android apps. The Upload service:

  * receives observation package (zip);

  * repairs faulty meta-data;

  * detects duplicate samples;

  * applies Data Quality Control module to extract processed FU value

  * uploads the meta-data to the PostgreSQL database, thus exposing it automatically through the GeoServer interfaces.

  Dependency: I.a, I.d, I.e

b. SendMail service

  i. It is used for sending user-uploaded transparency data to the scientists processing/working with the data.

  ii. It is used when app images are "flagged" as being "not water" or not a good measurement. In this case the image is marked in the database as potential bad measurement and at the same time the image is sent by email to experts for a second opinion. Dependency: I.a, I.d

**M. MERIS satellite services**

a. Raw satellite data: 300 GB of data of the North and Mediterranean Sea
Dependency: None

b. Satellite extraction services: web services for extracting specific temporal ranges / geo locations of the MERIS satellite data
Dependency: I.a, I.d, M.a
*Note: For each component, the infrastructure and other components needed have been specified under "Dependency". So, for example, to host Component M.b—Satellite extraction services on a machine, the following components are also needed*:

  • *I.a–A machine running Debian Linux 6.0.8 (Other Linux flavours are compatible.)*

  • *I.d–Python*

  • *M.a–The raw satellite data*

c. Satellite tile-map layer: satellite data accessible through the Geoserver
Dependency: I.a, I.c, M.a

**E. Citclops data explorer**
This is the interface for exploring, analysing and downloading all the Citclops data.

a. Citizen observations [http://www.eyeonwater.org/]: where a user can navigate through all the observation samples of seawater taken by users of the Citclops smartphone applications.
Dependency: I.a, I.c, M.a

b. Marine Data Analyser [http://citclops-data-explorer.herokuapp.com/marine-data-analyser]: where an advanced user (such as a decision maker or marine researcher) can observe more in-depth information on the state of seawaters such as the historical state of

bathing waters, seawater colour obtained from the MERIS satellite data and historic weather data.
Dependency: I.a, I.d, M.c (optional, for viewing satellite layer)

c. Marine Data Repository [http://www.citclops.eu/search/welcome.php].

**A. Citclops apps:**

a. Citclops colour apps for Android
Dependency: S.a, I.a, I.d, I.e

b. Citclops colour apps for iPhone
Dependency: S.a, I.a, I.d, I

## Migration of the Citclops data sexplorer

To migrate the Citclops Data Explorer from the project-based hosting system to a long-term one the following approach has been followed and is recommended for similar projects:

1. A static copy is kept of all Citclops data and code in an offline hard disk (at the coordinator's facilities).

2. Components S.a, S.b, E.a, A.a, A.b have been migrated to the EyeOnWater system maintained by MARIS and NIOZ.

3. Migration of MERIS satellite components M.a, M.b, M.c can be handled by organizations interested in these data and services.

4. Migration of component E.b (Citclops Data Explorer–Marine Data Analyser) can be handled by organizations interested in these services.

For each machine hosting one or more Citclops components, 10 GB of space are needed for the operating system and other infrastructure components. Apart from this, the only space-intensive component is M.a.—*Raw satellite data*, which will require an extra 300 GB on the machine that is hosting it. It is recommended to host all the components (including all MERIS components) on one machine.

To summarise, two machines have been prepared for migration that will persist post project (Each machine hosts the operating system and other infrastructure components.):

1. for hosting the MERIS components M.a, M.b, M.c (space required 310 GB);

2. for hosting the E.b—*Citclops Data Explorer–Marine Data Analyser* (space required 10 GB).

The permanent long-term home for Citclops component E.b—*Citclops Data Explorer–Marine Data Analyser* has been implemented and is accessible at Heroku, a virtual cloud platform, which offers a free virtual machine.

## EyeOnWater and Citclops's monitoring methodology after Citclops

The EyeOnWater concept provides good opportunities to be expanded and reused regionally, nationally or internationally. Examples of what could be done are:

1. *Expanding the coverage of parameters measured with the app, or making variations of the app*. Loading these data to the server would allow presenting them in various user interfaces and using them in decision support systems next to conventional monitoring.

2. *Combining remote sensing with in-situ sensing*. The system allows combining in-situ information coming from citizens or governmental monitoring stations with relevant satellite-based maps to help the validation and interpretation of the citizen-science measurements.

## Future updates

Two features to improve the functioning and usability of the EyeOnWater/Citclops platform has been defined, which can be useful to researchers wanting to build upon the current development, or to citizen-science projects developing their own platform:

1. With respect to how to colour the ocean and sea surface, there are two main alternatives: (a) the FU colour measured by the MERIS satellite services or (b) a satellite image. The first one is considered preferable because less distracting; the problems are

   i. that MERIS data are not available everywhere and consequently what to do where they are not available; and

   ii. what to do, in particular, in the transition areas between the two colouring methods. Also, the difference between contiguous FU numbers is very little and difficult to detect to the human eye.
   Alternative viewing modes could be:

   a. a 'true colour' mode that approaches the colour people see in the field; and

   b. a different scale converting the FU value into an indexed contrasting colour.

2. Possibility to add a grid (with coordinates).

## Conclusions and recommendations

The general conclusion of this research is that citizens are valuable contributors in quality and quantity to the objective of collecting, integrating and analysing large amounts of fragmented and diverse environmental data, to determine the current status and trends of the environment. The Citclops project, which was the context of this research, was carried out to contribute towards the achievement of this objective, to foster citizen science as a way to complement data collected in traditional ways by scientists and governments, and to involve citizens in scientific endeavours and decision making. The project's methods and tools were initially tested mainly in European waters, but there is no restriction to apply them globally. Citclops built on existing data bases and integrated relevant environmental data (the colour of the water, its transparency or clarity, and its fluorescence) from remote-sensing and in-situ observations, covering freshwater and marine habitats. A key feature of Citclops is the delivery of relevant, integrated information to multiple stakeholders and end users. Through the adoption and adaptation of existing standards, the integration with GEOSS, and the availability of its open-source software, Citclops enables greater interoperability of different data layers and systems, provides access to improved analytical tools and services, and contributes to harmonised environmental monitoring-schemes integrating citizen-science efforts, long-term research programs and mainstream data collection. Furthermore, Citclops can facilitate political decisions for sound environmental management, and help to protect the environment for human well-being at different levels, ranging from participatory park management to education about ocean health. Additionally, the project strengthens global capacities and infrastructure for environmental data and information management and sustainable development.

## Supporting information

**S1 Appendix. Required python libraries.**
(DOCX)

## Acknowledgments

The authors would like to thank all contributors from the Citclops's consortium: Eurecat, CSIC, UNIOL, NOVELTIS, TCD-Coastwatch, MARIS, NIOZ, Kinetical, Deltares, TriOS and Stichting VU/VUmc. Also, they express their gratitude towards 1000001 Labs, the members of Citclops's advisory board and the EC project-officer; and they thank Lyle Visa for his support in the final edits of the manuscript. They dedicate this publication to their co-author and friend Dr. Marcel Wernand, who passed away in 2018. His dedication to the study of the colour and clarity of the sea and his drive for citizen involvement was a great inspiration to the Citclops project and its participants.

## Author Contributions

**Conceptualization:** Luigi Ceccaroni, Jaume Piera, Marcel R. Wernand, Oliver Zielinski.

**Data curation:** Luigi Ceccaroni, Jaume Piera, Anna Friedrichs.

**Formal analysis:** Luigi Ceccaroni, Jaume Piera.

**Funding acquisition:** Luigi Ceccaroni, Jaume Piera, Marcel R. Wernand, Oliver Zielinski, Hendrik Jan Van Der Woerd.

**Investigation:** Luigi Ceccaroni, Jaume Piera, Anna Friedrichs.

**Methodology:** Luigi Ceccaroni, Jaume Piera, Oliver Zielinski, Hendrik Jan Van Der Woerd, Anna Friedrichs.

**Project administration:** Luigi Ceccaroni, Oliver Zielinski.

**Resources:** Luigi Ceccaroni, Jaume Piera, Oliver Zielinski.

**Software:** Luigi Ceccaroni, Jaume Piera, Hendrik Jan Van Der Woerd.

**Supervision:** Luigi Ceccaroni, Jaume Piera, Oliver Zielinski.

**Validation:** Luigi Ceccaroni, Jaume Piera, Hendrik Jan Van Der Woerd, Anna Friedrichs.

**Visualization:** Luigi Ceccaroni, Jaume Piera, Anna Friedrichs.

**Writing – original draft:** Luigi Ceccaroni, Jaume Piera, Marcel R. Wernand, Oliver Zielinski, Julia A. Busch, Hendrik Jan Van Der Woerd, Raul Bardaji, Anna Friedrichs, Stéfani Novoa, Peter Thijsse, Filip Velickovski, Karin Dubsky.

**Writing – review & editing:** Luigi Ceccaroni, Jaume Piera, Oliver Zielinski, Hendrik Jan Van Der Woerd, Anna Friedrichs, Meinte Blaas.

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
