## [Decision Letter · Decision Letter 0]

2 Jan 2020

PONE-D-19-29477

Citclops: A next-generation sensor system for the monitoring of natural waters and a citizens' observatory for the assessment of ecosystems’ status

PLOS ONE

Dear Dr. Ceccaroni,

Thank you for submitting your manuscript to PLOS ONE. After careful consideration, we feel that it has merit but does not fully meet PLOS ONE’s publication criteria as it currently stands. Therefore, we invite you to submit a revised version of the manuscript that addresses the points raised during the review process.

We would appreciate receiving your revised manuscript by Feb 16 2020 11:59PM. To enhance the reproducibility of your results, we recommend that if applicable you deposit your laboratory protocols in protocols.io, where a protocol can be assigned its own identifier (DOI) such that it can be cited independently in the future. For instructions see: http://journals.plos.org/plosone/s/submission-guidelines#loc-laboratory-protocols

We look forward to receiving your revised manuscript.

Kind regards,

Amitava Mukherjee, ME, Ph.D.

Academic Editor

PLOS ONE

Journal Requirements:

2. We note that Figures 7, 10, 22 and 26 include an image of a participant in the study. 

As per the PLOS ONE policy (http://journals.plos.org/plosone/s/submission-guidelines#loc-human-subjects-research) on papers that include identifying, or potentially identifying, information, the individual(s) or parent(s)/guardian(s) must be informed of the terms of the PLOS open-access (CC-BY) license and provide specific permission for publication of these details under the terms of this license. Please download the Connt Form for Publication in a PLOS Journal (http://journals.plos.org/plosone/s/file?id=8ce6/plos-consent-form-english.pdf). The signed consent form should not be submitted with the manuscript, but should be securely filed in the individual's case notes. Please amend the methods section and ethics statement of the manuscript to explicitly state that the patient/participant has provided consent for publication: “The individual in this manuscript has given written informed consent (as outlined in PLOS consent form) to publish these case details”.

3. We note that Figure 27 in your submission contains satellite images which may be copyrighted. All PLOS content is published under the Creative Commons Attribution License (CC BY 4.0), which means that the manuscript, images, and Supporting Information files will be freely available online, and any third party is permitted to access, download, copy, distribute, and use these materials in any way, even commercially, with proper attribution. For these reasons, we cannot publish previously copyrighted maps or satellite images created using proprietary data, such as Google software (Google Maps, Street View, and Earth). For more information, see our copyright guidelines: http://journals.plos.org/plosone/s/licenses-and-copyright.

You may seek permission from the original copyright holder of Figure 27 to publish the content specifically under the CC BY 4.0 license. 

If you are unable to obtain permission from the original copyright holder to publish these figures under the CC BY 4.0 license or if the copyright holder’s requirements are incompatible with the CC BY 4.0 license, please either i) remove the figure or ii) supply a replacement figure that complies with the CC BY 4.0 license. Please check copyright information on all replacement figures and update the figure caption with source information. If applicable, please specify in the figure caption text when a figure is similar but not identical to the original image and is therefore for illustrative purposes only.

We note that one or more of the authors are employed by a commercial company: MARIS BV.

5. Please ensure that you refer to Figures 21 and 25 in your text as, if accepted, production will need this reference to link the reader to the figure.

Reviewers' comments:

Reviewer's Responses to Questions

**Comments to the Author**

1. Is the manuscript technically sound, and do the data support the conclusions?

Reviewer #1: Yes

Reviewer #2: Yes

2. Has the statistical analysis been performed appropriately and rigorously? 

Reviewer #1: I Don't Know

Reviewer #2: Yes

3. Have the authors made all data underlying the findings in their manuscript fully available?

Reviewer #1: Yes

Reviewer #2: Yes

4. Is the manuscript presented in an intelligible fashion and written in standard English?

Reviewer #1: Yes

Reviewer #2: Yes

5. Review Comments to the Author

Reviewer #1: The authors have presented their Citclops project outcome in an self explanatory way. I appreciate their good work of the group. I have some concern regarding

1) What about interference study data,

2) interlaboratory comparison study of the data

3) What is the standard reference materials available in the study parameters, and how to define it?

4) Is there any standard reference materials developed for this kind of exploratory studies

5) Remote sensing interference data missing

6) While measuring color if fluorescence microbes are present it may interfere with the measurement? I don't the authors can explain it

7) How is the validation of these results done?

8) Supposing for a particular period of time if the data is not taken due to unforseen condition how the group work for missing link

Reviewer #2: The manuscript is well written and publishable in its current form. The authors are advised to re frame the abstract and introduction in little concise manner as in its current form it is to elaborative.

6. PLOS authors have the option to publish the peer review history of their article (what does this mean?). If published, this will include your full peer review and any attached files.

Reviewer #1: No

Reviewer #2: No

---

## [Author Response · Author response to Decision Letter 0]

18 Feb 2020

Dear editors,

This rebuttal letter responds to each point raised by the academic editor and reviewers. 

We think that the manuscript in its revised form meets PLOS ONE's style requirements .

2. We note that Figures 7, 10, 22 and 26 include an image of a participant in the study. As per the PLOS ONE policy (http://journals.plos.org/plosone/s/submission-guidelines#loc-human-subjects-research) on papers that include identifying, or potentially identifying, information, the individual(s) or parent(s)/guardian(s) must be informed of the terms of the PLOS open-access (CC-BY) license and provide specific permission for publication of these details under the terms of this license. Please download the Connt Form for Publication in a PLOS Journal (http://journals.plos.org/plosone/s/file?id=8ce6/plos-consent-form-english.pdf). The signed consent form should not be submitted with the manuscript, but should be securely filed in the individual's case notes. Please amend the methods section and ethics statement of the manuscript to explicitly state that the patient/participant has provided consent for publication: “The individual in this manuscript has given written informed consent (as outlined in PLOS consent form) to publish these case details”. If you are unable to obtain consent from the subject of the photograph, you will need to remove the figure and any other textual identifying information or case descriptions for this individual.

Figures 7, 10, 22 and 26 have been modified so that they now do not include images of participants.

3. We note that Figure 27 in your submission contains satellite images which may be copyrighted. All PLOS content is published under the Creative Commons Attribution License (CC BY 4.0), which means that the manuscript, images, and Supporting Information files will be freely available online, and any third party is permitted to access, download, copy, distribute, and use these materials in any way, even commercially, with proper attribution. For these reasons, we cannot publish previously copyrighted maps or satellite images created using proprietary data, such as Google software (Google Maps, Street View, and Earth). For more information, see our copyright guidelines: http://journals.plos.org/plosone/s/licenses-and-copyright.

Figure 27 has been removed.

4. Authors information. 

All authors information has been updated.

5. Funding Statement and Competing Interests Statement

These have been updated in the cover letter.

6. Please ensure that you refer to Figures 21 and 25 in your text as, if accepted, production will need this reference to link the reader to the figure.

We now refer to Figures 21 and 25 in the text.

Reviewer #1 concerns

7a. What about interference study data

7b. Remote sensing interference data missing

Although the term “interference” is not used in aquatic optics and instrument development, we understand that the reviewer is referring to a systematic analysis of potential disturbances in the observation. A report has been published on this topic (Van der Woerd et al., 2013). Here, the key elements of perfect observations of colour, fluorescence and transparency are described in protocols and the potential disturbances are summarized. Good practices to detect errors and correct them are described. This includes the remote sensing observations. We now refer to this document in the manuscript, in Section “Data-quality control”: “In science, the used methods are education, training and protocols, so that a scientist is properly trained to check in detail the conditions of an observation. Within Citclops the training, education and protocols were tested in multiple places with a large variety of people (see sections 6.4, 7.2 and 8.3), and a systematic analysis of potential disturbances in observation has been carried out (Van der Woerd et al., 2013). In this systematic analysis, the key elements of perfect observations of colour, fluorescence and transparency are described in protocols and the potential disturbances are summarised. Good practices to detect errors and correct them are described. This includes the remote sensing observations.”

8. interlaboratory comparison study of the data

9. What is the standard reference materials available in the study parameters, and how to define it?

10. Is there any standard reference materials developed for this kind of exploratory studies?

We have developed the new observation techniques described in the paper with the state-of-the art scientific insights and procedures. The idea behind this is that these citizen-science observations must have scientific merit. This goal was reached by detailed comparisons with standard laboratory equipment, field tests and publication in scientific journals. This is connection (A) in Figure 2 in the article By Busch et al. (2016a) (see below). For colour, the Forel-Ule scale was characterized in the laboratory (Novoa et al., 2013). The construction of a plastic scale was developed under strict standardized illuminations conditions and a detailed protocol for the construction of this reference material has been published (Novoa et al., 2014). For fluorescence, we used standard fluorophores and chlorophyll solution as reference. We now refer to this in the manuscript, in Section “Data-quality control”: “The new observation techniques described in this paper have been developed with the state-of-the art scientific insights and procedures. The idea behind this is that these citizen-science observations must have scientific merit. This goal was reached by detailed comparisons with standard laboratory equipment, field tests and publication in scientific journals. This is represented by connection “A” in Fig 2 of the paper by Busch et al. (2016a). For colour, the Forel-Ule scale was characterized in the laboratory (Novoa et al., 2013). The construction of a plastic scale was developed under strict standardized illuminations conditions and a detailed protocol for the construction of this reference material has been published (Novoa et al., 2014). For fluorescence, standard fluorophores and chlorophyll solution have been used as reference. The method investigated and used in this paper was tested in a Carl von Ossietzky Universitat Oldenburg’s laboratory in Wilhelmshaven. Field tests were performed at various sites. Additionally, two different scientific instruments were used to compare results: one instrument was a MicroFlu Chl-A from TriOS and the other one was a bench-top fluorescence spectrometer (LS55) from PerkinElmer. With respect to standard references, a dilution series of chlorophyll-a standard (5000 µg/L dissolved in 90% acetone) was used. No additional standard reference material was developed during test studies.”

11. While measuring color if fluorescence microbes are present it may interfere with the measurement? I don't the authors can explain it

The reviewer is correct that the spectral reflection of natural waters might be modified by the fluorescence signal of algae, coloured dissolved organic matter (CDOM) and microbes in the water. In natural waters, fluorescence by algae is a signal located near 685 nm and rather weak compared to the full algal absorption over the visual range. The quantum efficiency is of the order of 1%. Also, the colour matching function for the red band near 685 is very low, which implies that a small addition in reflection will have marginal impact on the red colour and thereby the observed FU scale. We added a sentence in Section “Colour” to explain that interference by fluorescence in the colour observation might be present, but is ignored for the moment: “Some natural phenomena can change water colour, and a particular colour does not necessarily mean that the water is of bad quality. For example, the spectral reflection of natural waters might be modified by the fluorescence signal of algae, CDOM and microbes in the water. In natural waters, fluorescence by algae is a signal located near 685 nm and rather weak compared to the full algal absorption over the visual range. The quantum efficiency is of the order of 1%. Also, the colour matching function for the red band near 685 nm is very low, which implies that a small addition in reflection will have marginal impact on the red colour and thereby the observed FU scale. Therefore, interference by fluorescence in the colour observation might be present, but is ignored for the moment.”

12. How is the validation of these results done?

We agree with the reviewer that an observation without validation is useless. For colour the following validation activities have taken place:

a. The FU scale as provided by the observer in the EoW app is compared to an FU scale that is derived with the WACODI (Novoa et al., 2015) algorithm at the server. This is described in detail in the manuscript. 

b. The WACODI algorithm was validated with state-of-the art above-water hyperspectral sensors in a field campaign that covered a wide range in natural waters, from lakes to oceanic waters. 

c. The protocols and algorithms to derive the FU scale from the MERIS satellite observations were validated with in-situ measurements. The results for many different natural waters are described in Wernand et al. (2013). A more extensive validation campaign in lakes near the Ebro has been published by Busch et al. (2016) 

For fluorescence measurement, validation was performed through parallel sampling and laboratory analysis following standard procedures at selected locations. Validation is explained in detail in Friedrichs et al. (2017a): laboratory experiments of the SmartFluo show a linear correlation (R2 = 0.98) to the chlorophyll-a concentrations measured by reference instruments, such as a high-performance benchtop laboratory fluorometer (LS 55, PerkinElmer).

These details about validation have been added in the corresponding sub-sections of section “Methods”.

13. Supposing for a particular period of time if the data is not taken due to unforseen condition how the group work for missing link

The following paragraph was added in section “Citizen water-colour monitoring apps” to respond to this comment: “Part of the impact of Citclops on science is based on the use of the "EyeOnWater-colour" app to collect water-quality data. This data collection can be interrupted for a number of reasons, the main two ones are presented hereafter together with the corresponding solution. For the cases in which there is an interruption in data uploading from smartphone to server due to connectivity reasons (for example because there is no mobile coverage where the observations are collected, such as in open ocean), there is a caching mechanism in place to ensure uploading of data stored locally on the client (smartphone) to the data server as soon as the phone is online. For the cases in which there is an interruption in data recording by citizens, quantitative matchups and trend analyses of citizen data as well as earth-observation data are predominantly carried out on aggregated datasets. Hence the focus is on larger-scale features in time and space than at the scale of the individual pixel or citizen recording. This upscaling reduces the impact of outliers and irregular gaps in time and space.”

Reviewer #2 concerns

14. The authors are advised to re frame the abstract and introduction in little concise manner as in its current form it is too elaborative.

The abstract and introduction have been reframed and are now more concise manner and less elaborative.

Best regards,

Luigi

---

## [Editor Report · Decision Letter 1]

21 Feb 2020

Citclops: A next-generation sensor system for the monitoring of natural waters and a citizens' observatory for the assessment of ecosystems’ status

PONE-D-19-29477R1

Dear Dr. Ceccaroni,

We are pleased to inform you that your manuscript has been judged scientifically suitable for publication and will be formally accepted for publication once it complies with all outstanding technical requirements.

With kind regards,

Amitava Mukherjee, ME, Ph.D.

Academic Editor

PLOS ONE
---

## [Editor Report · Acceptance letter]

25 Feb 2020

PONE-D-19-29477R1 

Citclops: A next-generation sensor system for the monitoring of natural waters and a citizens' observatory for the assessment of ecosystems’ status 

Dear Dr. Ceccaroni:

I am pleased to inform you that your manuscript has been deemed suitable for publication in PLOS ONE. Congratulations! Your manuscript is now with our production department. 

With kind regards,

on behalf of

Professor Dr. Amitava Mukherjee 

Academic Editor

PLOS ONE